# Cross-cultural adaptation and psychometric properties of the Italian version of the Body Perception Questionnaire

Francesco Cerritelli[1], Matteo Galli[1,2], Giacomo Consorti[1,3]*,
Giandomenico D'Alessandro[1], Jacek Kolacz[4,5], Stephen W. Porges[5,6]

1 Clinical-based Human Research Department, Foundation COME Collaboration, Pescara, Italy,
2 Research Department, SOMA, Istituto Osteopatia Milano, Milan, Italy, 3 Education Department of
Osteopathy, Istituto Superiore di Osteopatia, Milan, Italy, 4 Socioneural Physiology Laboratory, Kinsey
Institute, Indiana University, Bloomington, Indiana, United States of America, 5 Traumatic Stress Research
Consortium, Kinsey Institute, Indiana University, Bloomington, Indiana, United States of America,
6 Department of Psychiatry, University of North Carolina at Chapel Hill, Chapel Hill, North Carolina, United
States of America

☯ These authors contributed equally to this work.
* giacomo.consorti@gmail.com

pone.0251838

Hernandez de Elche, SPAIN

**Data Availability Statement:** All relevant data are
within the paper and its Supporting information
files.

## Abstract

### Background/Objective

The purpose of this study was to cross-culturally adapt the Body Perception Questionnaire
Short Form (BPQ-SF) into Italian and to assess its psychometric properties in a sample of
Italian subjects.

### Methods

A forward-backward method was used for translation. 493 adults were recruited for psycho-
metric analysis. Structural validity was assessed with confirmatory factor analysis and a
hypothesis testing approach. Internal consistency was assessed by Cronbach's alpha and
McDonald's omega. Measurement invariance analysis was applied with an age-matched
American sample.

### Results

The single-factor structure fit the awareness subscale (RMSEA = .036, CFI = .983, TLI =
.982). Autonomic reactivity (ANSR) was well-described by supra- and sub-diaphragmatic
subscales (RMSEA = .041, CFI = .984, TLI = .982). All subscales were positively correlated
(r range: .50-.56) and had good internal consistency (McDonald's Omega range: .86-.92,
Cronbach's alpha range: .88-.91). Measurement invariance analysis for the Awareness
model showed significant results (p<0.001) in each step (weak, strong and strict) whereas
the ANSR showed significant results (p<0.001) only for the strong and strict steps.

**Funding:** The authors received no specific funding for this work.

**Competing interests:** The authors have declared that no competing interests exist.

## Conclusions

Our results support the Italian version of the BPQ as having consistent psychometric properties in comparison with other languages.

## Introduction

The central nervous system is continuously updating the status of bodily states and visceral organs. The subjective experience of the ongoing bottom-up flow from the body may be clustered into a construct called "body awareness" (i.e. body perception) [1]. Research in the field of functional neuroanatomy, neurophysiology, and psychiatry has progressively uncovered the physiological and neural processes related to the subjective body experiences related to body awareness [2–5]. Interoception, the sense of the physiological condition of the body, is a neural process through which information from organs and tissues is transmitted to the brain, forming a neural pathway through which body awareness emerges. Incoming afferent information informs the functional regulation of tissues and organs mainly through the activity of autonomic nervous system (ANS) [1]. Consistent with an evolutionary perspective, interoception and ANS activity are crucial for preserving body homeostasis. Interoceptive and ANS central neural networks have been maintained and elaborated over the course of the evolutionary encephalization process, reaching higher-order cortical areas (e.g., anterior insula) where subjective or mental body awareness emerges. Mental awareness of body homeostasis improves homeostatic preservation controlling emotional behaviour and social communication [6].

Body awareness, also called body perception, is defined as the ability to recognize internal body cues [7] including changes of target organs innervated by the ANS (i.e., autonomic reactivity) [1]. Body perception has been found to be useful in the management of chronic diseases such as chronic low back pain [8,9], congestive heart failure [10], chronic renal failure [11,12], and irritable bowel syndrome [13–16]. Body awareness is also important for physical stability and wellbeing [8,10–12].

Several research disciplines, including psychiatry and somatic-oriented therapies, have shown interest in the body perception construct. Specifically, there is an interest in how indices of subjective body experience can complement laboratory-based measures [16]. Moreover, in many clinical settings patient subjective reports (e.g., pain and other subjective symptoms) are important sources of information that patients and clinicians can use to evaluate therapy progress and health status. Therefore, efforts have been made to create self-reported questionnaires to investigate body perception and autonomic reactivity, but few have shown strong psychometric properties with a theoretical coherence with the organization of peripheral neural pathways [1].

The Body Perception Questionnaire (BPQ) [17] is a self-reported questionnaire developed to assess the subjective experiences of the function and reactivity of target organs and structures that are innervated by the ANS. Its development has followed the theoretical division of the ANS described by the polyvagal theory [18]. The polyvagal theory is an evolutionary neurophysiological framework that divides the vagal circuits within the parasympathetic nervous system into a ventral vagal complex (VVC) and a dorsal vagal complex (DVC). VVC regulates the striated muscles of the face, head, and visceral organs above the diaphragm through efferent nerves that originate from the nucleus ambiguous in the brainstem; the physiological status of the VVC targeted organs is represented through sensory pathways that terminate in the nucleus of the solitary tract (NTS) in the brainstem. DVC efferent neurons originating in

the dorsal nucleus of the vagus regulate the organs below the diaphragm while the physiological status of these organs is represented in NTS through afferent vagal fibers.

The original BPQ was composed of 122 items, assessing body awareness, autonomic nervous system reactivity, cognitive-emotional-somatic stress response, body and cognitive stress response styles, and health history. Since its introduction, the BPQ has been frequently used in clinical research and translated into several languages [1,19]. However, recently a shorter version of the questionnaire was developed and validated, focusing primarily on two subscales: (1) Awareness (26 items); (2) Autonomic Nervous System Reactivity (ANSR) (20 items) [1].

The purpose of this study was to cross-culturally adapt the BPQ into Italian, to assess its psychometrics characteristics in a sample of Italian subjects, and to examine associations between the subscales of BPQ and the sample characteristics.

## Materials and methods

This psychometric study consisted of a cross-cultural adaptation and factor analysis of the Body Perception Questionnaire (BPQ). The study protocol was approved by the Internal Review Board of the Foundation Centre for Osteopathic Medicine Collaboration (COME IRB n.01/2019). All subjects gave their written consent and all procedures followed the Declaration of Helsinki.

### Participants

Four hundred and ninety three adults were recruited to complete the questionnaire during clinical visits with professional osteopaths. Therefore, the current sample might not be considered a regular community sample, but rather an osteopathic care sample. A comprehensive description of participants' characteristics is reported in Table 1. The target sample size was calculated taking into account at least 10 participants for each item of the questionnaire [20].

Data collection forms were designed at the Foundation COME Collaboration international coordinating center in Pescara, Italy. Participants completed the approximately 10-minute questionnaire online through the Google Forms platform, in the presence of the osteopath in their private practice. There was no financial incentive for completion.

Subjects reported socio-demographic and clinical characteristics, gender, age, height, weight, education level, work, annual income, medication usage, physical activity, smoking habits, pathologies. Questions concerning the medication use, physical activity and smoking habits allowed only dichotomous answers (yes/no) without quantitative specification, apart from the physical activity question (if ≥2/week). They also completed the Italian translation of the BPQ-SF (BPQ-I).

### The Body Perception Questionnaire (BPQ)

The Body Perception Questionnaire Short Form (BPQ-SF) is a self-report measure of the body awareness and experiences of autonomic reactivity [1,17]. It has demonstrated strong psychometric properties and a consistent factor structure across multiple languages [1,19]. It is composed of two domains: body awareness (26 items) and autonomic nervous system reactivity (ANSR; 20 items). The body awareness domain measures sensitivity to internal bodily functions (e.g. "During most situations I am aware of my mouth being dry."). The Autonomic Nervous System Reactivity (ANSR) domain is composed of a supradiaphragmatic reactivity subscale, which measures the typical experience of body reactions above the diaphragm (e.g. "During most situations I am aware of sweat in my armpits"), and a subdiaphragmatic reactivity subscale, which measures of gastrointestinal functions below the diaphragm (e.g. "During most situations I am constipated"). There is an item—"I feel like vomiting"—that is included

**Table 1. Sample characteristics.**

| Characteristics | Values |
|---|---|
| **Gender (%)** | |
| F: | 292 (59.23) |
| M: | 201 (40.77) |
| **Age (SD)** | 34.71 (14) |
| **BMI (SD)** | 23.46 (3.86) |
| **Education level (%)** | |
| High school diploma: | 247 (50.10) |
| University degree: | 200 (40.57) |
| **Other:** | 46 (9.33) |
| **Smoker (%)** | |
| yes: | 126 (25.56) |
| no: | 367 (74.44) |
| **Medications usage (%)** | |
| yes: | 119 (24.14) |
| no: | 374 (75.86) |
| **Psychiatric disorder (%)** | |
| yes: | 4 (0.81) |
| no: | 489 (99.19) |
| **Physical activity ≥2/week (%)** | |
| yes: | 289 (58.62) |
| no: | 204 (41.38) |
| **Other Diseases (%)** | |
| yes: | 146 (29.61) |
| no: | 347 (70.39) |

in both the supradiaphragmatic reactivity and the subdiaphragmatic reactivity domains. Responses measure frequency of sensations, assessed on a 5-point Likert-type scale ("Never" to "Always").

## Cross-cultural adaptation

The cross-cultural adaptation followed a previously used method for the adaptation of the same questionnaire in another language [1]. First, two native Italian speakers fluent in English independently translated the BPQ. One translator was an Italian professional with a medical background fluent in English. The other was a professional English-Italian translator with 20-year translation experience and no medical background. A common forward translated version was agreed upon by the two translators. Second, the provisional BPQ Italian version was independently back-translated by two English native speakers who were fluent in Italian. Third, all the translations and the provisional BPQ were discussed by an expert committee including the translators, a linguistic expert, two osteopaths, and one epidemiologist. The committee discussed the content by comparing semantic, idiomatic, experiential, and conceptual equivalence; the goal was to develop the pre-final Italian BPQ (BPQ-I) to be understandable to a reading level of a typical 18 years old. Fourth, the BPQ-I was pilot-tested with 20 healthy subjects (62% female; mean age 36.4 ± 6.6 years). After BPQ-I completion, subjects were asked to report any items with unclear meaning. All reported comprehension problems were discussed by three authors (FC, GC, GdA) and used to inform modifications of the BPQ-I.

## Statistics

Data were analyzed using different measures in relation to the type of data (continuous, ordinal, categorical and dichotomous). Mean, median, mode, point estimates, range, standard deviation and 95% confidence intervals were used. Data analysis was conducted using R version 3.5.1 (R Core Team 2017) and Rstudio Version 1.1.463 (RStudio, Inc 2009–2018). Hypothesis tests were conducted using a critical alpha value of 0.05.

## Data preparation

Confirmatory Factor Analysis (CFA) was performed on dichotomized items (0 = never, 1 = occasionally or more often) to maintain acceptable response cell sizes and replicate methods used with the BPQ in other languages [21].

## Psychometric assessment

The Consensus-based Standards for the selection of health Measurement Instruments (COSMIN) initiative [22] and the International Society for Quality of Life Research (ISOQOL) [23] were used as methodological guidelines. Validity refers to the degree to which a patient-reported instrument measures the construct(s) it purports to measure [23]. Two subdomains of validity were assessed in this study: structural validity, referred as the degree to which the scores of an instrument are an adequate reflection of the dimensionality of the construct to be measured, and construct validity/hypotheses testing, that is the degree to which the scores of an Health Related Patient-Reported Outcome instrument (HR-PRO) are consistent with hypotheses (for instance with regard to internal relationships, relationships to scores of other instruments, or differences between relevant groups) based on the assumption that the instrument validly measures the construct to be measured [1]. Exploratory factor analysis was computed on full-score items to assess the number of adequate factors, which was too elevated in previous studies and constrained Cabrera et al. to adopt dichotomization. Factor structure was assessed for convergence with previously reported dimensionality in English- and Spanish-language samples [1] using confirmatory factor analysis.

The hypotheses were that the age was normally distributed with respect to the Awareness subscale values and that age was negatively correlated with ANSR subscale value [24,25]. The hypothesis regarding physical activity was that it was positively correlated with Awareness and ANSR subscales values [26,27]. Furthermore, we hypothesized that Smoking habits [28], BMI [25], and educational level [29] are associated with the Awareness Subscale values.

## Factor Analysis and Measurement Invariance Analysis (MIA)

Confirmatory factor analysis was conducted using the R package "lavaan" [30]. Goodness of fit to the data was evaluated using root mean squared error of approximation (RMSEA), the Tucker-Lewis Index (TLI), and the Comparative Fit Index (CFI) [31–34]. We interpreted good fit to be evidenced by an RMSEA value near .06 or lower as well as CFI and TLI values near .95 or greater, as recommended by Hu and Bentler [35]. We used variance-covariance matrices, which dimensions corresponded the same as the items included per each model (Awareness model = 26x26; ANSR model = 20x20). Weighted Least Square Mean and Variance (WLSMV) was used as estimator according to Barendse *et al.* [36] that suggested using models with discrete responses [37]. Correlations between factors were not constrained, an analysis decision that can reproduce correlated or uncorrelated factor structures. Based on results from prior factor analysis on BPQ in other languages, CFA was performed to examine the fit of the (1) one-factor solution for the Body Awareness domain; (2) a two-factor solution for the

Autonomic Reactivity domain with supra-diaphragmatic and sub-diaphragmatic domains, uncorrelated each other. Based on the findings of Cabrera *et al.* [1] the item "I feel like vomiting" was included in both ASNR Supra and Sub-diaphragmatic subscale.

Furthermore, we used an age-matched American sample to assess measurement invariance of the factor structure between Italian and American responses. The American sample was recruited online and its collection and descriptive statistics are described in Cabrera *et al.* [1] study. To conduct the analysis, we used the *measurement Invariance* function of "semTools" R package [38], which performs multiple group analyses with increasing restrictions on parameters, from configural to strict invariance using the Chi-squared difference test, RMSEA, and CFI. Statistical significance of the Chi-squared difference test indicates that exact fit of the model has to be rejected. However, when sample size is large, small differences between observed and model-implied parameters can result in rejection of the model. To avoid hypothesis testing over-sensitivity due to high power, we also used the RMSEA, where values smaller than .05 indicate close fit, and values smaller than .08 are considered satisfactory. CFI values over .95 indicate also a reasonably good fit. CFI changes of .02 and RMSEA of .03 are most appropriate for tests of weak invariance with large group sizes, and variations of -.01 for ΔCFI and .01 for ΔRMSEA are appropriate for strong invariance tests. Partial invariance is evidenced if the majority of items on the factor are invariant [39].

### Internal consistency

Internal consistency is defined as the degree to which the items measure the same construct, based on their interrelatedness [22]. The Cronbach's alpha [40] and McDonald's omega were calculated using the R "psych" package [41]. McDonald's omega was selected because it showed to be a stronger index in case of categorical items and variable factor loadings as with BPQ-I items [42]. Internal consistency was computed for each subscale separately.

### Association of BPQ with demographic and clinical variables

Subscale scores were calculated and examined for association of demographic and clinical variables. Associations of BPQ scores with age and BMI were calculated using Kendall and Pearson's correlation coefficients. Categorical demographic variables were compared using Welch two sample t-tests.

In linear regressions we set each one with the respective subscales of the BPQ (Awareness, ANSR Supra and Sub-diaphragmatic scores) as the dependent variables and the demographic and clinical characteristics of the sample as independent variables (age, gender, physical activity and medication use), selecting the appropriate model using the stepwise methods. ANOVA was used to evaluate associations of the BPQ subscales with education.

### Floor and ceiling effects

Floor and ceiling effects were considered to be present if ≥ 15% of the patients reported the lowest (0) or highest (46 possible BPQ score. The effect was considered also for the BPQ subscales separately [43].

## Results

### Confirmatory Factor Analysis (CFA) and Measurement Invariance Analysis (MIA)

Confirmatory factor analysis showed that the single-factor structure fit the awareness subscale well (RMSEA = 0.036, CFI = 0.983, TLI = 0.982). Autonomic reactivity was well-described by

supra- and sub-diaphragmatic subscales (RMSEA = 0.041, CFI = 0.984, TLI = 0.982; Table 2). Supra- and sub-diaphragmatic reactivity were positively correlated (r = 0.56) and were respectively related to Awareness (r = 0.56, r = 0.50). All these correlations showed statistical significance (p< 0.001). To examine the possibility of other potential well-fitting factor structures, a post hoc exploratory factor analysis was performed on the raw (non-dichotomized items). The results did not reveal any novel well-fitting factor structure beyond that tested by the CFA (see supporting information).

Measurement invariance analysis showed that Chi-Squared difference test for Awareness model results significant in each step (weak, strong and strict) with p-values <0.001. Considering the RMSEA, values were 0.075 for the configural and weak invariance and 0.085 for the strong and strict ones. The CFI values were 0.85 for the configural and 0.84 for the weak invariance, 0.79 for the strong and 0.78 for the strict ones.

Analyzing the ANSR model, the "weak" step of Measurement Invariance was not significant (p = 0.55), whereas all the other steps (strong and strict invariances) showed significant values (p-value < 0.001). The RMSEA models produced values of 0.067 for the configural, 0.065 for the weak invariance, 0.069 for the strong and 0.071 for the strict one. The CFI values were 0.90 for the configural and for the weak invariance, .88 for the strong and for the strict ones. Table 3 reports results from Measurement Invariance Analysis.

## Descriptive statistics

The BPQ-SF was scored by adding the dichotomized responses (0 = never, 1 = occasionally or more often) in accordance to the factor structure described above. Descriptive statistics for the resulting scores are in Table 4. The sample mean score for the Awareness subscale was 19.3 (±6.05), for the supradiaphragmatic subscale of the ANSR was 6.3 (±4.41) and for the sub-diaphragmatic subscale of the ANSR was 3.71 (±1.96).

## Internal consistency

Results of McDonald's Omega were calculated for the Body Awareness subscale (0.92; CI: 0.90, 0.95), Supradiaphragmatic Reactivity (0.88; CI: 0.84, 0.90) and Subdiaphragmatic Reactivity (0.86; CI: 0.83, 0.89). Cronbach's Alpha results were also calculated for the Body Awareness subscale (0.91; CI: 0.90, 0.92), Supradiaphragmatic Reactivity (0.88; CI: 0.86, 0.90) and Sub-diaphragmatic Reactivity (0.78; CI: 0.75, 0.81).

## Associations with demographic variables

Correlations between BPQ-SF subscales with Age and BMI were calculated using Pearson and Kendall correlation coefficients (Table 5). Results showed negative correlations between age and BPQ-SF awareness and sub-diaphragmatic reactivity.

A linear regression model showed that males had lower awareness values compared to females (β = -1.28, CI: -2.39, -0.31; p = 0.02) and there was a significant negative association with age (β = -0.06, CI: -0.09, -0.02; p = 0.002; S2 Table in S1 File). ANSR supradiaphragmatic reactivity showed a negative association with age (β = -0.03, CI: -0.06, -0.004; p = 0.02), lower scores in males (β = -1.05, CI: -1.81, -0.29; p = 0.007) and lower scores for those who were physical active (β = -1.66, CI:-2.43, -0.89; p<0.001). A positive association was found with medication use (β = 0.97, CI:0.12, 1.82; p = 0.03; S3 Table in S1 File). The ANSR subdiaphragmatic reactivity linear regression model showed a negative association with age (β = -0.02, CI: -0.03, -0.005; p = 0.004), lower scores among males (β = -0.6, CI: -0.93, -0.26; p = 0.0005) and lower scores in those who were physical active (β = -0.44, CI: -0.79, -0.1; p = 0.01; S4 Table in S1 File).

**Table 2. Confirmatory Factor Analysis (CFA) response per items.**

| BPQ Items | Estimate | Standard Error | z Value | P |
|---|---|---|---|---|
| **Awareness** | | | | |
| "Swallowing frequently" | 1.000 | | | |
| "An urge to cough to clear my throat" | 1.026 | 0.110 | 9.345 | 0.000 |
| "My mouth being dry" | 1.107 | 0.125 | 8.829 | 0.000 |
| "How fast I am breathing" | 1.196 | 0.125 | 9.589 | 0.000 |
| "Watering or tearing of my eyes" | 1.190 | 0.122 | 9.763 | 0.000 |
| "Noises associated with my digestion" | 0.820 | 0.100 | 8.211 | 0.000 |
| "A swelling of my body or parts of my body" | 0.814 | 0.113 | 7.218 | 0.000 |
| "An urge to defecate" | 0.933 | 0.121 | 7.724 | 0.000 |
| "Muscle tension in my arms and legs" | 0.988 | 0.127 | 7.806 | 0.000 |
| "A bloated feeling because of water retention" | 1.032 | 0.129 | 8.006 | 0.000 |
| "Muscle tension in my face" | 1.400 | 0.146 | 9.584 | 0.000 |
| "Goose bumps" | 1.116 | 0.120 | 9.331 | 0.000 |
| "Stomach and gut pains" | 0.894 | 0.129 | 6.934 | 0.000 |
| "Stomach distension or bloatedness" | 1.011 | 0.127 | 7.988 | 0.000 |
| "Palms sweating" | 1.466 | 0.167 | 8.791 | 0.000 |
| "Sweat on my forehead" | 1.538 | 0.164 | 9.359 | 0.000 |
| "Tremor in my lips" | 1.655 | 0.183 | 9.048 | 0.000 |
| "Sweat in my armpits" | 0.947 | 0.113 | 8.374 | 0.000 |
| "The temperature of my face (especially my ears)" | 1.247 | 0.143 | 8.706 | 0.000 |
| "Grinding my teeth" | 1.182 | 0.152 | 7.771 | 0.000 |
| "General jitteriness" | 0.766 | 0.108 | 7.101 | 0.000 |
| "The hair on the back of my neck standing up" | 1.293 | 0.146 | 8.838 | 0.000 |
| "Difficulty in focusing" | 0.736 | 0.104 | 7.066 | 0.000 |
| "An urge to swallow" | 1.515 | 0.141 | 10.736 | 0.000 |
| "How hard my heart is beating" | 1.095 | 0.131 | 8.384 | 0.000 |
| "Feeling constipated" | 1.139 | 0134 | 8.488 | 0.000 |
| **ANSR Supradiaphragmatic** | | | | |
| "I have difficulty coordinating breathing and eating" | 1.000 | | | |
| "When I am eating, I have difficulty talking" | 0.938 | 0.074 | 12.602 | 0.000 |
| "My heart often beats irregularly" | 0.909 | 0.080 | 11.371 | 0.000 |
| "When I eat, food feels dry and sticks to my mouth and throat" | 0.995 | 0.073 | 13.692 | 0.000 |
| "I feel shortness of breath" | 0.975 | 0.081 | 12.095 | 0.000 |
| "I have difficulty coordinating breathing with talking" | 1.049 | 0.068 | 15.435 | 0.000 |
| "When I eat, I have difficulty coordinating swallowing, chewing, and/or sucking with breathing" | 1.012 | 0.063 | 16.160 | 0.000 |
| "I have a persistent cough that interferes with my talking and eating" | 0.879 | 0.072 | 12.131 | 0.000 |
| "I gag from the saliva in my mouth" | 1.061 | 0.071 | 15.012 | 0.000 |
| "I have chest pains" | 0.966 | 0.076 | 12.753 | 0.000 |
| "I gag when I eat" | 0.997 | 0.072 | 13.807 | 0.000 |
| "When I talk, I often feel I should cough or swallow the saliva in my mouth" | 1.071 | 0.080 | 13.336 | 0.000 |
| "When I breathe, I feel like I cannot get enough oxygen" | 1.077 | 0.077 | 13.908 | 0.000 |
| "I have difficulty controlling my eyes" | 0.848 | 0.077 | 11.017 | 0.000 |
| "I feel like vomiting" | 0.684 | 0.092 | 7.466 | 0.000 |
| ASNR Subdiaphragmatic | | | | |
| "I feel like vomiting" | 1.000 | | | 0.000 |
| "I have 'sour' stomach" | 1.963 | 0.392 | 5.013 | 0.000 |
| "I am constipated" | 2.097 | 0.407 | 5.151 | 0.000 |

*(Continued)*

**Table 2.** (Continued)

| BPQ Items | Estimate | Standard Error | z Value | P |
|---|---|---|---|---|
| "I have indigestion" | 2.400 | 0.456 | 5.265 | 0.000 |
| "After eating I have digestive problems" | 2.094 | 0.407 | 5.148 | 0.000 |
| "I have diarrhea" | 1.880 | 0.377 | 4.985 | 0.000 |

**Table 3.** Measurement invariance analysis per awareness and ANSR model.

| Steps | Awareness Model | | | ANSR Model | | |
|---|---|---|---|---|---|---|
| | CFI | RMSEA | $P(>|\chi^2|)$ | CFI | RMSEA | $P(>|\chi^2|)$ |
| Configural | 0.85 | 0.075 | - | 0.90 | 0.067 | - |
| Weak | 0.84 | 0.075 | 0.0001 | 0.90 | 0.065 | 0.55 |
| Strong | 0.79 | 0.085 | 0.0001 | 0.88 | 0.069 | 0.0001 |
| Strict | 0.78 | 0.085 | 0.0001 | 0.88 | 0.071 | 0.0001 |

Table shows the results of the Measurement Invariance Analysis for Awareness and ANSR models. The rows are the steps of testing invariance in order to increase constrains (from Configural to Strict). The columns show the values of *Comparative Fit Index (CFI)*, *Root Mean Squared Error of Approximation (RMSEA)* and of *P-value related to Chi square tests (P(>|χ2|))*.

## Floor and ceiling effects

Extreme floor or ceiling effects were not observed in our sample according to the $\geq$ 15% criterion. The lowest score (0) was presented in 2 subjects of our sample (0.41%), whereas the highest score (46) was reported in 26 subjects (5.27%).

**Table 4.** Descriptive statistics of BPQ-SF score.

| | Mean | Median | SD | Skew | Kurtosis | Min | Max |
|---|---|---|---|---|---|---|---|
| BPQ-SF Body Awareness | 19.3 | 21.0 | 6.05 | -0.90 | 0.30 | 0.0 | 26.0 |
| BPQ-SF Supradiaphragmatic Reactivity | 6.3 | 6.0 | 4.41 | 0.42 | -0.82 | 0.0 | 15.0 |
| BPQ-SF Subdiaphragmatic Reactivity | 3.7 | 4.0 | 1.96 | -0.497 | -0.98 | 0.0 | 6.0 |

**Table 5.** Pearson and Kendall's correlation indices.

| | Pearson | Kendall |
|---|---|---|
| Age and BPQ-SF Awareness | -0.0113 | -0.079 |
| Age and BPQ-SF ANSR | Supra: -0.037 | Supra: -0.078(*) |
| | Sub: -0.10(*) | Sub: -0.09(**) |
| BMI and Awareness | 0.014 | -0.004 |
| BMI and ANSR | Supra: -0.002 | Supra: -0.029 |
| | Sub: -0.025 | Sub: -0.062 |

(*) p-value <0.05;

(**) p-value<0.01;

(***) p-value<0.001.

Finally, an ANOVA model did not show association between education levels and the Awareness subscale score (F = 1.48; p = 0.22).

## Discussion

The aims of the present study were to adapt the BPQ into Italian language, to assess its psychometric characteristics in a sample of Italian subjects, and examine the associations between the subscales of BPQ and the sample characteristics. The confirmatory factor analysis showed comparable results to the English, Spanish, and Chinese versions previously described in the literature [1,19], with subscales reflecting body awareness, supradiaphragmatic autonomic reactivity, and subdiaphragmatic autonomic reactivity. All subscales also demonstrated strong internal consistency.

A post hoc exploratory factor analysis did not suggest an alternative factor structure different from that tested in this confirmatory factor analysis and previous studies. In the initial BPQ psychometric study [1], factor analysis on the full item distributions (5 ordered categories using polychoric correlations) required the estimation of an excess number of parameters with initial results suggesting that the full response item loadings were resulting in overfitting of the data and large influence of random noise. The solution employed by those authors was to create binary cut offs that would be less sensitive to noise and overfitting the data, which produced reliable, interpretable factor solutions that could be replicated across samples. However, this left unresolved questions about whether the factor structure could be replicated with the full item distributions. In this study, the post-hoc EFA conducted on the full-item distributions did not reveal a better fitting factor structure and thus supported the factor structure that has been previously described.

The results of the current analysis demonstrated partial measurement invariance between the two BPQ subscales of the English and the Italian version. In order to understand the possible reasons for this partial invariance, methodological and cultural factors need to be considered. It has been argued that MIA is influenced by the sample and model size. Putnick and Bornstein [39] affirmed that in large samples (N> 100) the chi-squared test increases its power to reject the null hypothesis. In the case of the present study, therefore, the total sample was 1009 (Italian = 493, U.S.A. = 516), supporting the assertion that statistical power was very high and capable of identifying small, possibly insubstantial differences. Another important factor that influences the MIA is model size. According to Putnick and Bornstein [39], smaller models (e.g. composed by 4 factors with 2 indicators) are more likely to show more sensitive CFI and RMSEA indices than larger models (e.g. 4 factors with 6 indicators or larger). Considering the model size of the current research (Awareness = 26 items; ANSR = 20 items), it is evident that the MIA is characterized by a large model. As a consequence, the CFI and RMSEA indices might be influenced by this methodological element and therefore the results need to be considered in relation to the model size.

It is worth noting that comparing two samples from different countries might lead to additional bias. Indeed, another factor to explain the partial invariance should be sought in the cultural differences present in the two populations. Differences possibly present also in the reading and understanding of the BPQ-SF items, despite the fact that the translation from English to Italian followed the WHO international guidelines [44]. For these reasons, we argue that the results of the MIA likely partially reflect the cross-cultural adaptation of the Italian version of the BPQ-SF. Moreover, since the study did not include convergent or discriminant validity testing the study can only be used to inform understanding of the dimensionality of the Italian BPQ-SF.

Negative associations were found between ANSR subscale and age, physical activity, and male gender. Similarly, a negative association between age and male gender and the awareness subscale was observed. Positive associations were found between medication use and ANSR supradiaphragmatic subscale (S3 Table in S1 File). Our findings are in accord with Cabrera

*et al.* [1] and with several previous works [24,45] which found a decrease of interoception and cardiac autonomic regulation by the ventral vagal complex [25,46,47] in elderly people. Physical activity seems to have no influence on the Awareness subscale, which is not consistent with other studies using different instruments of body awareness [26,27]. This difference might depend on the instrument used for the measurement. In fact, Multidimensional Assessment of Interoceptive Awareness is a questionnaire that measures interoception using a distinct framework from the BPQ. Some authors describe interoception as encompassing three distinct dimensions (accuracy, sensibility, and awareness) [48,49]. According to these authors, the BPQ Body Awareness subscale measures interoceptive sensibility. The different facets of the same construct measured could justify the observed difference between our result and other studies [26,49]. Since some of our findings differ from the prior literature relating to physical activity, further investigations are needed to examine relations between physical activity and Awareness and ANSR. However, one potential hypothesis to explain the absence of link between physical activity and body awareness might be based on reflecting a 'body numbness', that is a feeling of harmless sensation, which has been shown to be associated with those who exercise as their primary mode to regulate bodily state [29].

Smoking was not associated with differences in any BPQ subscales. Our results are not consistent with the hypothesis proposed by Naqvi and Bechara [28]: the interpretation framework between the awareness of the interoception and smoking takes into account that the insula—which can be affected by nicotine making the interoceptive information available to conscious awareness. Since the role of the insula in processes related to conscious interoception is established [50,51], the observed correlation did not converge with prior evidence and theory.

In general, our results showed a significant difference on ANSR supradiaphragmatic score between participants who are using medication compared to participants who are not. This finding is in line with previous studies [52] showing interoceptive alteration following medication usage. All the subscale scores were higher in women, compared to men. This finding is consistent with Cabrera *et al.* [1] suggesting robustness to cultural and language differences of the BPQ [46,53]. The results of the Welch two sample t-test between gender showed a significant difference among Male and Female groups in all BPQ subscales, consistent with those of Antelmi *et al.* study [46].

## Limitations and recommendations for future studies

The present study includes some limitations. The psychometric evaluation of the questionnaire did not encompass the assessment of convergent and divergent validity. Therefore, it is not possible to affirm that the BPQ-I actually measures the intended constructs (i.e. body awareness and autonomic reactivity). Although the body awareness construct has been studied in previous studies [49], the autonomic reactivity construct still needs to be tested with sensor-based measures. Further studies are needed to better define those constructs and their properties. Participants were recruited from a pool of clients seeking osteopathic care. Therefore, the current sample is not a regular community sample, but rather an osteopathic care sample, where a significant percentage of subjects reported a health-related condition. It is possible that participants in this sample may have higher autonomic reactivity or lower awareness as part of their reason for seeking care. Further studies need to be conducted with more general samples to establish better normative data and replicate the psychometric features of the scale. Furthermore, the participants in the sample were highly educated (40% with a university degree). A previous study underlined how education level might impact the body awareness of subjects with a direct correlation [54]. However, the sample in the study was specific and

therefore the results might be difficult to generalize. In order to understand the relationship between education level and body awareness further studies are needed.

## Conclusions

Our results support the Italian version of the BPQ as having consistent psychometric properties in comparison with other languages. Applying the BPQ-I might help to identify the status of individual circuits that contribute to dysfunction and the development of novel interventions that can target autonomic or interoceptive dysfunction [1].

## Supporting information

**S1 File.**
(DOCX)

## Author Contributions

**Conceptualization:** Francesco Cerritelli, Giacomo Consorti, Giandomenico D'Alessandro, Jacek Kolacz, Stephen W. Porges.

**Data curation:** Francesco Cerritelli, Matteo Galli, Jacek Kolacz, Stephen W. Porges.

**Formal analysis:** Francesco Cerritelli, Matteo Galli, Giacomo Consorti.

**Investigation:** Francesco Cerritelli, Giacomo Consorti, Giandomenico D'Alessandro.

**Methodology:** Francesco Cerritelli, Giacomo Consorti, Giandomenico D'Alessandro, Jacek Kolacz, Stephen W. Porges.

**Project administration:** Francesco Cerritelli.

**Resources:** Francesco Cerritelli.

**Supervision:** Francesco Cerritelli, Jacek Kolacz, Stephen W. Porges.

**Writing – original draft:** Francesco Cerritelli, Matteo Galli, Giacomo Consorti, Giandomenico D'Alessandro.

**Writing – review & editing:** Francesco Cerritelli, Matteo Galli, Giacomo Consorti, Giandomenico D'Alessandro, Jacek Kolacz, Stephen W. Porges.

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
