## [Decision Letter · Decision Letter 0]

11 Jun 2020

PONE-D-19-35072

Cross-cultural adaptation and validity of the Italian version of the Body Perception Questionnaire.

PLOS ONE

Dear Dr. Consorti,

Thank you for submitting your manuscript to PLOS ONE. After careful consideration, we feel that it has merit but does not fully meet PLOS ONE’s publication criteria as it currently stands. Therefore, we invite you to submit a revised version of the manuscript that addresses the points raised during the review process.

We look forward to receiving your revised manuscript.

Kind regards,

Antonio Palazón-Bru, PhD

Academic Editor

PLOS ONE

Journal Requirements:

2. We note you have included a table to which you do not refer in the text of your manuscript. Please ensure that you refer to Table 2 in your text; if accepted, production will need this reference to link the reader to the Table.

Reviewers' comments:

Reviewer's Responses to Questions

**Comments to the Author**

1. Is the manuscript technically sound, and do the data support the conclusions?

Reviewer #1: Partly

2. Has the statistical analysis been performed appropriately and rigorously? 

Reviewer #1: I Don't Know

3. Have the authors made all data underlying the findings in their manuscript fully available?

Reviewer #1: Yes

4. Is the manuscript presented in an intelligible fashion and written in standard English?

Reviewer #1: No

5. Review Comments to the Author

Reviewer #1: My review exceeded 20.000 characters. I upload a file that includes my General Comments about the manuscript, the several major issues I believe should be revised as well as list with several minor issues for the authors to resolve.

6. PLOS authors have the option to publish the peer review history of their article (what does this mean?). If published, this will include your full peer review and any attached files.

Reviewer #1: No

---

## [Author Response · Author response to Decision Letter 0]

9 Jul 2020

Review of the manuscript

Manuscript number PONE-D-19-35072, entitled “Cross-cultural adaptation and psychometric properties of the Italian version of the Body Perception Questionnaire.”

Dear editor,

Dear reviewers,

We greatly appreciate your readiness to have read our paper and to provide us with relevant feedback and useful suggestions to further improve the quality of our paper. A detailed description of all changes has been provided below.

For any further information, please do not hesitate to contact us.

Editor

and

Response: Thank you. Done.

2. We note you have included a table to which you do not refer in the text of your manuscript. Please ensure that you refer to Table 2 in your text; if accepted, production will need this reference to link the reader to the Table.

Response: Thank you. Done.

Response: Thank you. Done.

Reviewer 1

1. Define study goals more clearly

1.1. After reading the manuscript it seemed to me that this study had three main goals:

- Cross-cultural adaptation of the BPQ Short-Form for Italian.

- Examine the psychometric properties of the Italian BPQ Short-Form

- Examine the association between BPQ subscales and sociodemographic/clinical variables (e.g. age, BMI, smoking, diagnosis, medication intake, etc).

These goals should be stated at the end of the Introduction section and clearly noticed by the reader across the manuscript.

Response: Dear Reviewer, we appreciate your suggestion. The addition of main goals at the end of the introduction will make the reading of the whole paper more clear. 

1.2. Clinimetric vs Psychometric: I know this is just a terminology issue, but I was surprised when the authors described their study as clinimetric. The terms are widely overlapping (De Vet et al. 2003 - 10.1016/j.jclinepi.2003.08.010) and some argue that clinimetrics is not a unique approach, but is rather a subset of psychometrics (Streiner, 2003 - 10.1016/j.jclinepi.2003.08.011). I believe that this term is not very frequently used nowadays (a quick search on PubMed records in the last 5 years confirmed this). I would suggest that the authors use the term psychometric study as it is much more widely used.

Response: Dear Reviewer, the decision of using the term clinimetrics derived from De Vet et al. 2011 (Measurement in medicine: a practical guide) and from Feinstein 1987 (Clinimetrics). According to these authors we used clinimetrics as ‘measurement of clinical phenomena’. They focused on the construction of clinical indexes, and promoted the use of clinical expertise, rather than statistical techniques, to develop measurement instruments” (Feinstein 1987). We know that the terms are widely overlapping, and that clinimetris is not frequently used; for this reason, we accept your advice preferring Psychometric.

Reviewer 1

2. Improve the Introduction section: Although the introduction provides valuable information about body awareness and how this concept is linked to other phenoms (e.g. mental illness, management of chronic diseases), I still felt that this section was a little bit confusing and that some information was lacking.

Response: Dear Reviewer, thank you for your suggestions. We added the lacking informations and we structured the Introduction section as you suggest in the following points.

2.1. Although I know that the terminology in this field is quite heterogeneous, it would be important for me to clearly understand whether the authors consider body awareness and body perception similar constructs. The authors kind of state that at line 61 (page 3), but if the authors truly consider them as similar this should be explicitly stated at the start of the Introduction section. I actually believe it would be easier simply to talk about body awareness for consistency purposes, especially because it is one of the BPQ subscales.

Response: Thank you for your advice, we clarified better the constructs.

2.2. The introduction was mainly focused on defining and debating body awareness. This allows the reader to understand the body awareness domain, but not the autonomic nervous system reactivity subscale. Actually, I felt that this subscale was not evidently explained across the whole manuscript. The authors should clearly define both body awareness and automatic reactivity. I do not think this is an issue as the BPQ subscales are proposed to assess different domains and not a single unitary construct.

Response: Thank you! Done!

2.3. The authors introduced the concept of interoception and its association with body awareness and rightfully so in my opinion, as the BPQ is widely used as a measure for interoception in behavioral sciences. However, I believe the authors should further explore the most recently debated models about interoception when addressing this topic. This might seem unnecessary or not valuable for the main goals of this work, but in my opinion, this discussion is key when addressing the validity of this scale (especially for the body awareness subscale) and could help to interpret the results obtained by the authors. (more information below). If the authors feel that this information is not required in the Introduction, this should at least be debated in the Discussion section. Please check these three overviews about interoception measurement (the main ones that I know of in the last 5 years).

- Garfinkel et al. (2015) - 10.1016/j.biopsycho.2014.11.004

Argues that interoception encompasses three distinct dimensions (accuracy, sensibility, and awareness). According to these authors, the BPQ Body Awareness subscale measures interoceptive sensibility.

- Khalsa et al. (2017) – 10.1016/j.bpsc.2017.12.004

Proposed that interoceptive awareness encompasses a complex taxonomy (attention, detection, magnitude, discrimination, accuracy, sensibility, insight, self-report). This work also provides a lot of information about the link between interoception and mental health which I think is relevant for this manuscript.

- Murphy et al. (2019) – 10.3758/s13423-019-01632-7

Proposes a 2 x 2 factorial structure of interoception (factor 1: what is measured – accuracy vs. attention; factor 2: how is it measures – beliefs vs. performance). According to the authors, BPQ measures beliefs about interoceptive attention.

Response Thank you for your suggestion. The proposed literature has been added in the discussion

2.4. The authors decided to differentiate body awareness from somatosensory amplification. Although this is interesting from a conceptual standpoint, I am not sure if it really had a lot of valuable information as the authors do not address this concept in any other section.

Response: Dear Reviewer, as you rightly suggest, we have decided to omit the concept of somatosensory amplification to make the introductory section clearer.

2.5. I think that the structure of the introduction should be rethought for the purposes of clarity. The section is not very fluid. Reorganizing the several pieces while using some subheadings will likely make it substantially better and easier to follow. Thus, a possible structure would be (although there are other alternatives that may better suit the authors):

- Define body awareness (including its link to interoception) and autonomic reactivity;

- Describe the physiological circuits of body awareness and autonomic reactivity (including the polyvagal theory);

- Describe the importance of body awareness and autonomic reactivity for clinical disorders (e.g. mental health, other chronic disorders, etc) and for patient subjective reports;

- Highlight the lack of studies addressing the psychometric proprieties of measures related to body awareness and automatic reactivity. Introduce the BPQ.

- Study Goals

Response: Dear Reviewer, thank you for your suggestions. We improve the structure of the introduction as you request. 

Reviewer 1

3. Organization of Methods Section

I do not think the Methods section is very well organized. The authors seem to go back and forth between the different stages of the study. Maybe try to follow something like:

- Participants (only describe participant recruitment, sample size, and participant characteristics)

- Measures

- BPQ: Start by describing the BPQ more clearly. This is not done in any section of the manuscript. For instance, the authors never mention that this questionnaire is completed through a 5-level Likert scale that rates frequency. 

- Describe demographic and clinical data collected. End this section by provided information about the questionnaire application online

- Cross-Cultural Adaptation

- Statistical Procedures (provide an initial overview of statistics -info in lines 173-178 - before describing each specific step)

- Data Preparation

- Factor Analysis and Measurement Invariance Analysis

- Internal Consistency

- Association between BPQ and demographic/clinical variables

- Floor and ceiling effects

Response: Dear Reviewer, in order to improve the organization of this section, we accept your advices as possible to set “Methods” as you suggest. The description of BPQ can be found on the original validation paper. Therefore we would avoid to weightening the manuscript with those information.

Reviewer 1

4. Sample Characteristics

4.1. The authors state that they are assessing the BPQ validity in a sample of healthy subjects. However, I could easily argue that this not very accurate. First, all participants were contacted during clinical appointments with osteopaths, most of them probably due to some health-related condition. Secondly, around 30% of your sample reported some sort of disease and almost 25% reported medication intake. Although there is no specific information about the number of subjects per type of disease (this should also be added, at least in the supplemental materials), I think that considering this sample “healthy” is a little bit farfetched. I do not have any clear suggestions for the correct terminology that the authors should use, but clinical-sample or sample of health care services users would be more accurate in my opinion.

Response: Dear Reviewer, as well noted, we erroneously wrote in the abstract and in the introduction that our study was conducted on healthy subjects. As pointed out by your careful review, we have recruited subjects afferent to osteopathic care, and probably that may influence the selection of sample. If it is correct, we should expect a high presence of pathologies related to the osteopathic profession, which affect the musculoskeletal or neurological system. Musculoskeletal pathologies were not present in the questionnaire administered as they are very common. If we observe the number of participants by pathology - present in the table added on your suggestion - we observe that only about 5.5% of the total reports "other pathologies". We can assume that this percentage also contains musculoskeletal pathologies, which added to the percentage of subjects with neurological pathologies reaches about 8%. This fact does not make our sample composed of healthy subjects, but it reduces the influence of the selection mode. In conclusion we decided to substitute “healty subjects” with “Italian subjects”

4.2. The recruited sample is quite well-educated (40% with a university degree). Furthermore, about 30% of the participants had no income. These subjects with no income where mostly university students or just unemployed/retired? This should also be clear in my opinion as it can also strengthen the fact that this sample had above-average education levels. High education levels may be a possible source of bias that should be briefly introduced when describing study limitations.

Response: Dear Reviewer, observing the mean age (34) of our sample we can suppose that these subjects with no income were mostly university students or young degreed people looking for a job. This scenario is plausible with the Italian socio-economic situation.

Reviewer 1

5. Limitations of Online Surveys Should Be Addressed

5.1. The authors state the questionnaire was administered online, but there is not any more information about this. The authors should provide information on factors that are known to influence data quality in online surveys such as: average time of completion (probably not an issue in this study but should be reported); incentive for study completion (I guess there was not but this should be stated); missing values (most online studies do not have this issue but this should be stated as well); other factors the authors consider can influence data quality.

Response: Dear Reviewer, we really appreciate your suggestion and we added this information in our manuscript.

5.2. The authors did not describe any strategy to ensure data quality and exclude inadequate respondents. This may seem a minor issue, but there is evidence suggesting that careless respondents can range between 3 to 60% in self-report studies. Nowadays there are several methods to address this issue (for more information check Curran 2016 - 10.1016/j.jesp.2015.07.006 - and Niessen et al. 2016 - 10.1016/j.jrp.2016.04.010), but I recommend that the authors at least exclude subjects based on response time. Niessen et al. (2016) reported that response time was one of the most effective methods to detect careless responding. When subjects take more time to complete the questionnaire, data quality is better and there is less probability of careless responding. The authors should establish a cut off criteria to exclude subjects based on response time. There are also several suggestions about which criteria to use (e.g. three-second-per-item rule, exclude subjects 3 standard deviations below the average response time, etc) and the authors should select the criteria they feel it is more accurate (and justify it with previous studies). If the authors have not recorded response times, please look at other statistical procedures (e.g. multivariate outlier analysis). The authors may also check if there were answers submitted from duplicate IP addresses, although in this case, this seems to be highly unlikely.

Response : Dear Reviewer, unfortunately we did not recorded time of BPQ compilation, the use of Google Form returned us only the timestamp when sending reply. 

5.3. Finally, the limitations of online surveys should be addressed in the Discussion section. Although participants were directly contacted, I am assuming that only subjects with internet access could participate (unless the authors had any solution to address this). This introduces a slight bias that should be referred (for instance, older participants are more likely to have no internet access).

Response: Dear Reviewer, acute observation. Of sure internet connection was needed, but the direct contact with the recrutator, often in a Wi-Fi open zone, may eased the subministation of BPQ.

Reviewer 1

6. Issues with Psychometric Analysis

Although the psychometric analysis is quite interesting, I would like to clarify several issues with the authors:

6.1. Construct Validity: The authors argue that they will achieve construct validity through hypothesis testing and propose some hypotheses in the Methods sections without providing previous evidence for them. This evidence should be quite strong and from several sources ideally. I would like the authors to argue whether there is clear evidence about the relationship between body awareness and automatic reactivity with age, but more importantly to the other proposed variables such as smoking, physical activity, BMI, and educational level. Furthermore, why did the authors did not include gender in this hypothesis testing as they address it on the Results and the Discussion section? Regardless, I do not believe that this strategy is sufficient for construct validity, especially pertaining to body awareness as there are so many authors arguing how interoception encompasses several measurement dimensions (as described above). This is the main issue of this work in my opinion. For me, the only alternative to accomplish this would be to examine convergent and/or discriminant validity. For instance, Cabrera et al. (2017) used the Stress Reactivity Index and SomatoSensory Amplification Scale for convergent validity. As several authors also consider that the body awareness subscale assesses interoceptive sensibility or interoceptive attention, another alternative would be using interoception questionnaires to test for convergent/discriminant validity. Alexithymia could also be measured together with interoception as a recent meta-analysis from Trevisan et al. (2019) found that the BPQ was positively associated with alexithymia, although other measures of interoception were negatively associated (10.1037/abn0000454). Finally, depression and anxiety scales could also be an interesting alternative as they have been differentially associated with interoceptive scales, including the BPQ (e.g. Murphy et al. 2019 - 10.1177/1747021819879826). There should also be several other interesting alternatives for convergent or discriminant validity regarding the automatic reactivity subscale. Basically, if the authors want to dramatically improve their work and provide clear evidence that the Italian BPQ is truly measuring body awareness and automatic reactivity is to complete another study with a smaller sample size (I would estimate about 100 although a power analysis would be needed). If this is not possible for the authors, then they should acknowledge that this work is extremely limited to provide evidence for construct validity-and provide recommendations for future studies to test this.

Response: Thank you for your very insightful comment. The construct validity was based on the COSMIN framework, which provided the following elements for the construct validity (including structural validity, hypotheses testing, and cross‐cultural validity - https://www.cosmin.nl/wp-content/uploads/COSMIN-methodology-for-content-validity-user-manual-v1.pdf ). So following this framework, we believe that the methods used fulfilled the COSMIN criteria. 

Although we recognised the importance of comparing the BPQ with other measurements (which might be referred to as convergent or divergent validity), due to time constraints to resubmit the manuscript, we preferred to include those elements suggested in the limitations of the study. 

6.2. Although it is not my core expertise, I have some questions to be clarified/improved by the authors regarding the Confirmatory Factor Analysis (CFA) and Measurement invariance analysis (MIA). First, the description of the CFA procedures in the Methods section seems too broad. Further details should be given to the reader (e.g. type of matrix used, estimation procedure, etc), at least in Supplemental material. Second, the indicators chosen for Table 2 regarding the CFA are not very usual. If I understood correctly, the estimates are the unstandardized factor loadings, with their corresponding standard errors. Why did the authors select not to report the standardized loadings (more common) as it was done for instance by Cabrera et al. (2017)? Is it actually necessary to report “P(>|z|)| as the values are similar for each item? For more information about reporting a CFA I advise the authors to consult the work from Jackson et al. 2009 - 10.1037/a0014694. Third, I wanted to know why the authors selected to report the Chi-Square Statistics for the MIA but not for the CFA. Conversely, why did the authors report the TLI for the CFA but not for the MIA. This should be explained to the reader.

Response: Dear Reviewer, thank you for your comments. The decision of the indicators is referred to Hu and Bentler “Cutoff criteria for fit indexes in covariance structure analysis: Conventional criteria versus new alternatives. Structural Equation Modeling. 1999;6(1):1–55”. The P(>|z|) = 0.000 shows the significance of the CFA on the model, we think that is important for the reader. The choice to report these values for MIA and CFA are to be found in the sources of Hu and Bentler (1999) and Putnick and Bornstein “Measurement invariance conventions and reporting: The state of the art and future directions for psychological research. Vol. 41, Developmental Review. Mosby Inc.; 2016. p. 71–90”. 

6.3. In lines 181-183 the authors state that they dichotomized ratings as it was done on previous psychometric analysis of the BPQ (I believe the authors are refereeing to the paper from Cabrera et al. 2017, although the reference numbers seem incorrect). This is a valid argument, however, in the work from Cabrera et al. 2017 the authors actually decided to dichotomize the items as the exploratory factor analysis using full item distributions resulted in a large number of factors and loadings with complex factors. Did the authors perform any sort of factor analysis for the full-item distribution in this proposed Italian version? If so, this should be addressed somewhere (even if it is in the supplements). If not, I believe it will be important for the authors to complete this analysis. If the questionnaire was applied with a 5-point Likert scale, I believe that this should not be simply left out because it can provide valuable information for future studies.

Response: Dear Reviewer, we really appreciate your careful comment. In this study we decide to follow the same process of validation used from Cabrera 2017 that dichotomized data to perform EFA. With the same their reason -large number of factors and loadings with complex factors- we performed CFA only on the dichotomized data. While the BPQ is on a 5-point Liker type scale, but we think was important to follow the procedures of previous papers. 

6.4. Item “I feel like vomiting”: The authors only address the fact that this item was included on both reactivity subdomains in the Descriptive Statistics heading. This should be already explained in the Methods sections. Also, I think the authors should justify why did they select to keep the item on both subdomains In the work of Cabrera et al. (2017) the supra- and subdiaphragmatic reactivity loadings were a problem in the exploratory factor analysis of the Spanish sample. However, they also described that in the CFA factor loading of the American sample the item was associated more strongly with subdiaphragmatic reactivity. Thus, I think the authors should clearly explain why this item was retained on both subdomains in the Italian version.

Response: Dear Reviewer, we really appreciate your constructive revision. As you suppose we select to keep the item in both subdomains according to Cabrera et al 2017. In order to clarify this choice we decided to explicit it in the Methods section. 

6.5. The authors did not report the correlations between the supra- and sub-diaphragmatic reactivity scores with the body awareness score. This should be added.

Response: Thank you for your suggestion, we added the correlation between reactivity scores and awareness score.

6.6. In the internal consistency subheading, the authors should also report the confidence intervals for McDonald’s omega.

Response: Thank you. Done.

Reviewer 1

7. Testing the association between BPQ and demographic/clinical variables.

7.1. In the methods section (lines 217-220, page 10) the authors should state that subscale scores were calculated based on dichotomized items. Here the authors should also briefly describe which regression models were tested (how many, dependent, and independent variables).

Response: Thank you. Done.

7.2. I would also like to know why the authors decided to use both Pearson and Kendall correlations, which are parametric and non-parametric, respectively. Was that the reason? If so, did the authors did any kind of assumption testing before deciding to select both?

Response:Dear Reviewer, thank you for your comment. We decided to perform both Pearson and Kendall correlation not considering parametric and non-parametric features of the tests. Kendall rank test is more statistical correct to assess correlation in Likert scales (Arndt et al. 1999). However, Kendall tau seems to be stronger in a small/moderate sample, therefore we decided to compare it with the Pearson test.

7.3. In the results section, the authors describe the association between the BPQ subscales and each demographic/clinical variable. This is quite confusing for the reader. I do not understand if they prepared a regression model for each predictor or if they had a complex model with all the predictors. For the purpose of clarity, I think the authors should organize this section by first describing the significant correlations found (based on Table 6) and only them report the results of regression models (assuming the authors had one model with all predictors).

\\

Response: Thank you! Done

7.4. Table 6 should be presented as a correlation matrix. The way it is presented now is quite confusing. Statistically significant correlations should be highlighted in some way.

Response: Thank you! Done!

7.5. The authors reported correlations for (supposedly) categorical variables such as Smoking and Physical Activity. Computing correlations is not very advisable here in my opinion. The more adequate statistical approach would be to compare Smoker vs. Non-Smokers or Physical Active vs. Inactive using independent samples t-tests.

Response: Dear Reviewer, we complete the analysis suggested and we add them to the correlation computed. 

7.6. I think that Figure 1 is not necessary as this correlation could be merely reported in Table 6 and in the text.

Response: Dear Reviewer, thank you for your comment, we added the Fig. 1 to visually represent the the correlation, but if you think it superfluous, we remove it. 

7.7. I have several issues with the linear models sent in the supplementary material, which are actually described in the manuscript. First, I do not understand why the authors computed a regression model with the BPQ total score. The authors do not talk about a BPQ total score across the whole manuscript and I do not think the BPQ was designed to have a total score. At line 300, page 15 the authors also address this model. This analysis should not be reported in my opinion. Secondly, the authors presented linear models with more than 20 predictors. I do not think that the sample size is adequate for this analysis. Third, the authors do not report how many subjects have each pathology. For instance, if you include psychiatric disorders as a predictor, there are only 4 subjects with a diagnosis. This cannot provide valuable insights in a regression model. This issue is probably true for other diagnoses, but the authors must clarify how many subjects are in each type of pathology. If the authors are interested in exploring the association between BPQ and diagnosis, maybe it would be easier to perform independent group t-tests for each diagnostic group (only for groups with significant sample size) and report mean and standard deviation for each group. You can even control the p-value for multiple comparisons. Mann–Whitney U tests may also be used if the sample size is reduced and/or normality assumption is not met

Response: Dear Reviewer, thank you for your comment. In order to your suggestion we remove the results of the BPQ total score.We also added the numbers of each pathology in supplementary materials. However, in the regression model used pathologies are considered as an independent factor, which is weighted within the model, for this reason we think that the use of these analyses is adequate. 

7.8. At line 311, Page 16 the authors describe an ANOVA analysis that was never mentioned in the Methods sections.

Response: Thank you! Done

Reviewer 1

8. Enhance the Discussion section

8.1. The first paragraph (lines 321-323, page 16) should reinforce the 3 main goals that I argued at the beginning of my review. As it is now written it seems to empathize only internal consistency which is only a part of the authors work.

Response: thank you for your comment. Done.

8.2. Please debate the results following the same order of the results section for the purposes of consistency. Start by addressing the factor structure, followed by the internal consistency, and ending with the association with demographic/clinical variables.

Response: Thank you for your comment. Done

8.3. Between lines 336 and 343 (page 18), the argument is really confusing. “that is 26 and 20 factors for Awareness and ANSR”. Do the authors mean 2 factors with 26 and 20 items respectively? I had a hard time understand this argument so please rephrase this paragraph.

Response: Thank you for your comment. The paragraph has been rephrased.

8.4. Although the statistical analysis is not my core expertise, the way the authors described their results from the MIA actually led me to think they are worse than they actually are. As I understood correctly, the authors would consider RMSEA values of .08 as satisfactory. Looking at Table 3 one can see that almost all values are bellow .08. Only the Strong and Strict steps had an RMSEA of .085 which is still quite satisfactory in my opinion. I think the authors should emphasize this more in both the Results and Discussion section.

Response: Thank you! We add the explanation of why we intend the MIA as partial. 

8.5. The paragraph between lines 344 – 348 (page 18) addresses the main issue of this work, which is construct validity as I previously described. Although the translation process applied by the authors was outstanding, this work does not provide any data or analysis that allows them to truly state that the instrument is measuring body awareness and automatic reactivity. If the authors are not able to complete any sort of convergent/discriminant validity, this limitation should be explicitly debated here.

Response: Thank you for your comment. Unfortunately we are not able to provide the requested analysis. Therefore, we added the following sentence: “Moreover, since it was not planned a convergent/discriminant validity testing it might be difficult to assess to what extent the Italian version of the BPQ measures body awareness and automatic reactivity.”

8.6. Although there is a clear need for convergent validity testing in my opinion, the authors should also clearly highlight that some expected associations with demographic variables were retained in the Italian sample, which suggests, as the authors frame it and very well, “solid robustness to cultural differences”. Thus, I would make a paragraph highlighting this right after the paragraph about the Measurement Invariance Analysis. Only after addressing this topic, I would talk about any other associations (physical activity, smoking, etc).

Response: Thank you for your suggestion. Done. We added the recall to the figure 1 to ease the comprehension

8.7. Please consider my comments above about correlation analysis for smoking and physical activities. If the authors want to debate this association, they should revise the statistical analysis as suggested.

Response: Thank you. Done.

8.8. After revising the statistical analysis for physical activity, if the author's results are retained, they should explore why their results are different from the previous study. The most likely reason, in my opinion, is due to the instruments used to assess interoceptive awareness in previous studies (reference 34 and 35). For instance, one of these studies uses the Multidimensional Assessment of Interoceptive Awareness which is a questionnaire that measures interoception with a very different framework from the BPQ.

Response: Thank you for your comment. We added the literature you suggested in a comment above to discuss the tool/related difference.

8.9. Please improve the explanation about the association between body awareness, interoception, smoking, and interoception. I believe I understood what the authors meant, but the text is very unclear

Response: Thank you for the comment. The sentence has been rephrased as follow: “ A possible explanation for the correlation is proposed by Naqvi & Bechara (41): the interpretation framework between the awareness of the interoception and smoking takes into account that the insula - which can be affected by nicotine - makesing the interoceptive information available to conscious awareness.”

8.10. As I argued in previous comments, I highly question the regression models computed to test the associations with pathology. Thus, the conclusion drawn between lines 379 – 389 should be removed if the authors do not address these previous issues (possibly replacing regression by t-tests), including describing how many subjects there were for each disease category.

Response: Again, thank you for your comment. As above mentioned we added the numbers of each pathology in order to have more comprehensive information. However, we believe that the regression model used can be considered adequate as pathologies are considered as an independent factor, which is weighted within the model. Besides, as the total sample is sufficient for justifying the multivariate analysis based on all the factors included, we believe that the Student t-tests might produce misleading and less informative results.

8.11. At Line 385, Page 19 the authors state “a positive correlation has been found between ANSR supradiaphragmatic and psychiatric disease” but this is not described anywhere else, including the supplemental material. Furthermore, there were only 4 subjects with psychiatric disorders, which would not allow the authors to draw this conclusion. Please clarify this.

Response: Thank you for your accurate reading. We corrected the wrong sentence adding the following sentence: “Our findings showed a positive correlation between ANSR Subdiaphragmatic subscale and total score and the use of psychoactive drugs.”

8.12. The Discussion section is clearly lacking a paragraph describing study limitations as there are many. I addressed several limitations across this review, and they should be included somewhere in the Discussion section, probably at the end.

Response: Thank you for your suggestion. We added a sentence about limitations after each section of the discussion to keep it contextualized as much as possible.

8.13. In the Conclusion section, the authors may need to rephrase the sentence stating that this study supports the validity for measurement of body awareness and automatic reactivity if you do not address the issues about convergent validity.

Response: Please refer to the comments 6.1 

Reviewer 1

9. Inconsistencies, References Issues, and Written Expression

Here I report the main inconsistencies, reference misplacement, and written expression problems that I detected, although there are several other similar issues that should be revised. I truly recommend a full document revision to make it easier for the reader.

9.1. Line 85, Page 4: The references for the BPQ and polyvagal theory are incorrect (20, 21, and 22).

Response: Thank you. Done.

Line 114, Page 5: Here the authors describe reference 1 as their guidelines for cross-cultural adaption. Then, in the Discussion section (line 348, page17) they state that the translation followed the WHO international guidelines (33). Furthermore, in the reference list, there is a paper about cross-cultural adaption guidelines (20) that is not included in the Methods sections. Please clarify this.

Response: Thank you for your comment. Reference 1 refers to the statistical and conceptual methods. Reference 33 refers to the sole translation process. Reference 20 has been changed. However, the numbers of the references will change in the present version because we added some new ones and changed the order of others.

9.2. Line 137-138, Page 6: Not sure what the authors meant by “standard forms” and “consensus methodologies”

Response: Thank you for your comment. It has been rephrased.

9.3. Line 141-142, Page 6: The authors should be clearer when they state they recorded data about physical activity, smoking habits, pathologies, and drug usage. For instance, participants were asked how many times they performed physical activity each week? Did they describe which activities or how long? Regarding smoking, it was just a yes or no question? Did the authors ask how many cigarettes? Regarding pathologies, was it an open-ended question and the type of pathology was categorized after data collection? The same for drug usage.

Response: Thank you for your comment. Most of the answers to your questions can be found in Table 1. Since the manuscript is almost around 5000 words we would avoid to add more in text content to avoid making it too heavy.

9.4. Line 155-157: The definition of construct validity is quite confusing. Do not know what the authors mean by “HR-PRO instrument”. Written expression is also not particularly clear.

Response: Thank you, “HR-PRO instrument” means “Health Related Patient-Reported Outcome”. We added the extended expression in the manuscript.

9.5. Line 207, Page 10: Simply name this subheading “Internal consistency”

Response: Thank you! Done!

9.6. Line 212-213, Page 10: “However” does not really make sense in this sentence. It should be something like “We selected McDonald’s omega because ….”

Response: Thank you. Done.

9.7. Line 216, Page 10: I do not think that this subheading should be named “Reliability”. I would simply call it something like “Association of BPQ with demographic and clinical variables”.

Response: Thank you. Done.

9.8. Line 222, Page 10: Any reference to justify your cut-off point for floor/ceiling effects?

Response: Dear Reviewer, we add this reference : ” Terwee et all. (2007). Quality criteria were proposed for measurement properties of health status questionnaires. Journal of clinical epidemiology, 60(1), 34-42.”

9.9. Line 235, Page 11: Please report the p-value for the r = .56 correlation.

Response: Thank you! Done!

9.10. Lines 271-273, Page 15: The reporting of confidence intervals for Cronbach’s Alpha is not very clear.

Response: Thank you. Done.

9.11. Line 331, Page 17: Incorrect reference

Response: Thank you. Done.

9.12. Lines 365-267, Page 18: Poorly written sentence

Response: Thank you. Done.

9.13. Line 375, page 18: "The linear regression confirmed all the correlations and highlighted new ones." Please be very careful with terminology regarding statistical findings. This sentence is incorrect. Also, please avoid using the term "correlated" when talking about regression results. Rather used "positively associated" or "negatively associated".

Response: Thank you. Done.

9.14. Table 1: Replace Height and Weight by BMI as this variable was the one used for regression models. I would remove Annual Income from this table as it is not used again for any analysis. As previously stated, I think it would be more important to report unemployment status for the reader to understand how many were university students, employed workers, etc. Finally, Sports should be replaced by Physical Activity as these concepts do have the same meaning. From what I understood, the authors assessed Physical Activity.

Response: Thank you! Done!

9.14. Tables 2 and 3: These tables are not mentioned in the text.

Response: Thank you! Done!

9.15. Table 2: I would add subtitles to the table to identify the reported statistics (e.g. P(>|z|)

Response: Thank you! Done!

9.16. Table 5: I did not feel that this table was necessary (just repeated data on the text).

Response: Thank you. Table deleted.

Sincerely,

The authors

---

## [Decision Letter · Decision Letter 1]

31 Jul 2020

PONE-D-19-35072R1

Cross-cultural adaptation and psychometric properties of the Italian version of the Body Perception Questionnaire.

PLOS ONE

Dear Dr. Consorti,

Thank you for submitting your manuscript to PLOS ONE. After careful consideration, we feel that it has merit but does not fully meet PLOS ONE’s publication criteria as it currently stands. Therefore, we invite you to submit a revised version of the manuscript that addresses the points raised during the review process.

We look forward to receiving your revised manuscript.

Kind regards,

Antonio Palazón-Bru, PhD

Academic Editor

PLOS ONE

Reviewers' comments:

Reviewer's Responses to Questions

**Comments to the Author**

1. If the authors have adequately addressed your comments raised in a previous round of review and you feel that this manuscript is now acceptable for publication, you may indicate that here to bypass the “Comments to the Author” section, enter your conflict of interest statement in the “Confidential to Editor” section, and submit your "Accept" recommendation.

Reviewer #1: (No Response)

2. Is the manuscript technically sound, and do the data support the conclusions?

Reviewer #1: Partly

3. Has the statistical analysis been performed appropriately and rigorously? 

Reviewer #1: No

4. Have the authors made all data underlying the findings in their manuscript fully available?

Reviewer #1: Yes

5. Is the manuscript presented in an intelligible fashion and written in standard English?

Reviewer #1: No

6. Review Comments to the Author

Reviewer #1: The manuscript was fairly improved, mainly in the Introduction and Discussion section. However, in my opinion there are still several issues that need to be address before considering this manuscript for publication:

- BPQ Total Score

In their response letter the authors state that they excluded the results from the BPQ Total Score. However, they still address the total score in several sections of the manuscript:

Page 10: “The Cronbach’s alpha and McDonald’s omega were calculated for the total scale and after removing each item to evaluate this measurement property.”

Page 10: “In the four linear regressions we set, each one with the respective subscales of the BPQ (Awareness ANSR Supra and Sub diaphragmatic and BPQ total score) …”

Page 16: “In these cases we identified a positive significant correlation with BPQ total score and the presence of Cranio-Facial Pathologies …”.

Page 20: “Our findings showed a positive correlation between ANSR Subdiaphragmatic subscale and total score and the use of psychoactive drugs.”

The authors should remove all references to the total score if they truly excluded these analysis from their work.

- Binary Scoring System

In my first revisions, I asked why the authors decided to use the binary scoring system for the factor analysis. The authors argued decided that they decide to replicate the previous work. However, I still think the authors should provide alternative results for the full-scoring system, even if these results are displayed in the supplementary materials. The full-scoring system for the BPQ provides much more sensitivity for individual differences and should be used rather than the binary scoring system.

Thus, the authors should at least present an EFA with the full-scoring ratings. This is valuable information for the field as it would allow to understand if in the Italian the full-scoring rating also leads to a large number of factors and loadings with complex factors.

- Statistical Analysis and Reporting

1) The CFA procedures described in the Methods section is still quite incomplete. Further details should be given to the reader (e.g. type of matrix used, estimation procedure, are factors allowed to correlate or not, etc), at least in Supplemental materials.

2) Correlations with dichotomous variables: Even after the first revisions, the authors still describe and discuss correlations between scores and dichotomous variables like Smoking, Physical Activity or Drug Usage. As I argued before, correlations are not well suited for dichotomous variables. Correlations should be used with continuous variables such as Age or BMI. For dichotomous variables (e.g. Smoking, Physical Activity, Drug Usage, etc), the authors should only report independent samples t-tests comparing both groups (e.g. Drug Users vs. Non-Drug Users).

3) Regression models: I still do not clearly understand which regression models were implemented. More particularly, it is not clear which predictors were used. It seems that age, gender, physical activity, pathologies and drugs were used in the models presented in the supplementary tables. Smoking and BMI were not included in these models, apparently. Why? And psychiatric disorders were not included as well! This predictor is an important clinical outcome. Why was is not included?

More importantly, I cannot agree with the authors argument for conducting the models with all the pathologies and drugs as predictors. First, the authors presented linear models with more than 20 predictors. The sample size is not adequate for that many predictors. Secondly, there are predictors which do not have a sufficient amount of subjects per category to allow them to be included in the model.

For instance, there is only 1 subject that reported Immunosuppressor drug use. How can this be a predictor in the model? The same for cancer (n=2) and haematological disorders(n=3)? I do not understand the argument that “used pathologies are considered as an independent factor, which is weighted within the model”. The problem in the analysis is that these models include several dichotomous predictors which lack an adequate number of subjects per category. This is inadequate in my opinion, regardless of the independent factor argument.

4) Reporting of statistical results is not very adequate and does not follow any guidelines. The authors should at least report data for statistical tests using the APA guidelines.

- Limitations and Recommendations for future studies

The authors stated that they added a sentence about limitation after each discussion section. However, I still feel that there the manuscript lacks a clear paragraph describing limitations and recommendations for future studies. For instance, the authors do not debate the limitations of their sampling procedures. Collecting data only from subjects looking for osteopathic care is a limitation of this study. Furthermore, this sample is quite well-educated (40% with a university degree), which further limits the conclusions of this work. Also, the authors could also provide clear suggestions for future studies that would allow for convergent validity.

- Written Expression and Citations

Written expression is still quite flawed across the document. Several sentences are extremely hard to understand. To be considered for publication, this manuscript should be revised by someone fluent in English writing. There are also still a few issues with citation. For instance, the work form Cabrera et al. (1) is cited across the text in sentences where it is not adequate in my opinion. This should also be addressed.

Bellow I also provide specific comments for each section of the manuscript:

Introduction

This section has improved significantly since the last review. The constructs (and how they are linked) are more plainly defined. The text is also much more readable and clear for the reader. There is still one minor issue to be addressed.

- Page 3, 2nd paragraph: In my opinion, if body awareness equates to body perception in the authors’ opinion, this concept should be introduced in the first paragraph: “…. Clustered into a construct called body awareness (also known as body perception)”.

Furthermore, the definition of body awareness provided in the 2nd paragraph should be moved somewhere in the first paragraph. It does not make much sense to present the definition only in the 2nd paragraph as the construct was already introduced to the reader in the previous paragraph.

Methods

Page 5

- In authors reply to my original review, they state: “… we have recruited subjects afferent to osteopathic care, and probably that may influence the selection of sample. If it is correct, we should expect a high presence of pathologies related to the osteopathic profession, which affect the musculoskeletal or neurological system.” – I believe this issue should be addressed in the Participants heading.

- “The total number of participants was calculated taking into account at least 10 participants for each item of the questionnaire”. Unusual phrasing to explain sample size calculation. Please improve.

- The questionnaires were completed “in the presence of the osteopath” but it still not clear where. Private practices of each osteopath?

- “The use of this compilation method …”. Not sure what the authors means by “compilation method”.

Page 6

- Incomplete information on sociodemographic and clinical variables. For instance, regarding physical activity you should at least state that you asked whether subjects performed physical activity ≥2 times/week. Regarding smoking, it should be clear that it was a yes and no question. “Smoking habits” leads readers to wonder whether you asked how many cigarettes per day or per week.

- “It has demonstrated strong psychometric properties and a consistent factor structure across multiple languages”. – This is somewhat of an overstatement. To my knowledge, the BPQ was only formally valeted in English and Spanish.

- The BPQ description is still far too incomplete for a validation paper in my opinion. Although some information can be found in the Results section, I believe that this should be presented together with the scale description. The authors should describe how many items are included in each domain. Also, there is an item that is included in both the supradiaphragmatic reactivity and the subdiaphragmatic reactivity domains. This should be transparent for the reader. It should also be clear that separate scores are completed for the Body Awareness and Autonomic Reactivity domains (the BPQ manual does not even address a total score).

Page 7

- “The process of cross-cultural adaptation followed established guidelines 1.” – This citation does not seem right as the work from Cabrera does not provide any sort of guidelines for cross-cultural adaptation.

Page 8

- Again, citation number 1 (Cabrera et al.) seem to be inadequately used to explain structural validity and construct validity. Please use adequate citations to justify your rationale for psychometric analysis.

Page 9

- “As suggested by the Consensus-based Standards for the selection of health Measurement Instruments (COSMIN) initiative 23 and the International Society for Quality of Life Research (ISOQOL) 22” – This sentence seems incomplete.

- “The hypotheses were that the age was normally distributed with respect to the Awareness subscale values ….” – I could not understand what the authors meant with this sentence.

- “Furthermore, we hypothesized that Smoking habits, BMI and educational level with the Awareness Subscale values.” – This sentence seems incomplete or incorrect.

- “CFA was performed considering the subscales of BPQ, the Awareness and ANSR Supra and Sub-diaphragmatic.” – It is not clear that you conducted two separate CFA: 1) a one-factor solution for the Body Awareness domain; 2) a two-factor solution for the Autonomic Reactivity domain. Please improve this.

Page 10

- “The Cronbach’s alpha and McDonald’s omega were calculated for the total scale and after removing each item to evaluate this measurement property. We evaluated the internal consistency related to BPQ Awareness, BPQ ANSR total, and in the supra/subdiaphragmatic subscales.”

This is extremely confusing. First, the authors state that they computed the internal consistency for the total scale, which does not make much sense as there is no formal recommendation to use the total score for the BPQ. Then, I do not understand what the authors meant when they state “… after removing each item to evaluate this measurement property.”. Please clarify.

- I think the authors should describe cut-off scores to classify internal consistency scores.

- “The study of the correlations among BPQ scores and demographic characteristics was performed using Linear Regression Model, and estimating Kendall and Pearson’s indicators.” – Extremely unusual written expression. Please improve.

- “Studying Smoke and Physical activity variables, we computed t-test between smokers and no smokers, and between active and inactive subjects, in order to assess differences in Awareness and ANSR subscales” – Somewhat confusing sentence. Please improve.

- “In the four linear regressions we set, each one with the respective subscales of the BPQ (Awareness ANSR Supra and Sub diaphragmatic and BPQ total score)” – Still wondering why a regression model was computed for the BPQ Total Score. What does this BPQ Total Score measure actually, according to the authors? For me it seems that the BPQ subscales address different constructs (although they can be related with each other). So, if the authors use the BPQ total score they should clearly define what this total score represents.

“ … as the dependent variables and the demographic and clinical characteristics of the sample as independent variables.” – It is still not clear which variables were included as predictors. This is not clearly described on any part of the manuscript.

Page 11

- “Floor and ceiling effects were considered to be present if ≥ 15% of the patients reported the lowest

(0) or highest (47) possible BPQ score” – The BPQ Short Form only was 46 items. So the total score should only be 46, using the binary scoring system. If the authors considered the max score of 47, this is because they probably included item #41 (“I feel like vomiting”) twice, because this item is included on both the Supra and Sub Diaphragmatic domains. This would be incorrect in my opinion. Regardless, I do not think the authors should even consider the total score for analysis. However, if they decide to do so, this issue should be addressed.

Results

Page 14

- “The BPQ‐ SF was scored by adding the dichotomized responses (0 = never, 1 = occasionally or more often) in accord to the factor structure described above, with “I feel like vomiting” included in both reactivity scales (supra and subdiaphragmatic subscales).” – In my opinion, this information should be in the Methods section, rather than here.

Page 15, 16 and 17

I still find the description of the results of the section “Relation to demographic variables” quite confusing, for several reasons:

1. Why did you retain the correlation between Physical Activity and Smoking Habits? This variables are dichotomous correct? Thus, only the t-test should be used, not the correlation analysis.

2. The authors stated that they removed the analysis for the Total score in their reply to the first revision. However, they still report the regression model for the total score. Please clarify this.

3. The results are presented without any clear structure and it very hard to follow them. For instance, I still don’t understand clearly which variables were included in the regression models. In the results section, may be it would be easier to report the correlation, t-tests and ANOVA first. Them, prepare one short paragraph for each regression model, so that the reader can clearly understand the results.

4. Statistical reporting is quite poor. The authors should consider the APA guidelines for reporting their results.

Page 16

- I still think that Figure 1 should be removed. It’s not useful to interpret the study results.

Page 17

- “Floor or Ceiling effects were not present in our sample considering the ≥ 15% of the patients

reported the lowest (0) or highest (47) possible BPQ score” – Improve written expression.

Discussion

Page 18

- “For these reasons, we argue that the results of the MIA influence only partially the cross-cultural

adaptation and the validity of the Italian version of the BPQ-SF” – Improve written expression

- “The internal consistency showed to be very high for both subscales of the BPQ, and for the two

subscales of ANSR (Table 5).” – To make this statement the authors should have defined cut-off criteria for the internal consistency measures. Please address this.

- “Moreover, since it was not planned a convergent/discriminant validity testing it might be difficult to assess to what extent the Italian version of the BPQ measures body awareness and automatic reactivity”

After this statement, I think the authors should address how this is particularly true for the BPQ Body Awareness subscale and there is a lot of debate regarding the different measurement models of interception. I know the authors introduce this issue subsequently, but this seems the ideal spot to introduce this issue.

Page 19

- “However, there is some evidence in contrast with the negative association we observed between physical activity and ANSR 37. Physical activity correlated negatively with the subscale Awareness, which is not consistent with other studies using different instruments of body awareness 37,38” – Two sentences addressing the same argument. Please revise.

- “In fact, Multidimensional Assessment of Interoceptive Awareness which is a questionnaire that measures interoception with a very different framework from the BPQ.” – This sentence is incomplete.

- “Interestingly being a smoker seems to correlate with a heightened awareness on the BPQ awareness subscale.” – Correlate may not be the most adequate term. Also written expression should be improved in this sentence.

- “ … the interpretation framework between the awareness of the interoception and smoking takes into account that the insula - which can be affected by nicotine making the interoceptive information available to conscious awareness.” – After reading it multiple times, I still cannot understand this sentence. Please revise it.

Page 20

- “This result might be correlated with the findings from the study of Danner et al 46 in which the highest awareness …” – Again, correlated is not the most adequate term.

- “Our findings showed a positive correlation between ANSR Subdiaphragmatic subscale and total score and the use of psychoactive drugs. This finding is in line with previous studies 49” – Again, the authors stated in the reply to the revision that the total score would be removed, however they still debate it in the Discussion section. Furthermore, I do not understand how can the authors report a correlation with the use of psychoactive drugs, which is (if I understood correctly) a dichotomous variable. At the most the authors could compare drug users and non-users with an independent samples t-test.

7. PLOS authors have the option to publish the peer review history of their article (what does this mean?). If published, this will include your full peer review and any attached files.

Reviewer #1: No

---

## [Author Response · Author response to Decision Letter 1]

9 Oct 2020

Review of the manuscript

Manuscript number PONE-D-19-35072, entitled “Cross-cultural adaptation and validity of the Italian version of the Body Perception Questionnaire.”

Dear editor,

Dear reviewer,

We greatly appreciate your readiness to have read our paper and to provide us with relevant feedback and useful suggestions to further improve the quality of our paper. A detailed description of all changes has been provided below.

For any further information, please do not hesitate to contact us.

Reviewer 1

- BPQ Total Score

In their response letter the authors state that they excluded the results from the BPQ Total Score. However, they still address the total score in several sections of the manuscript:

Page 10: “The Cronbach’s alpha and McDonald’s omega were calculated for the total scale and after removing each item to evaluate this measurement property.”

Page 10: “In the four linear regressions we set, each one with the respective subscales of the BPQ (Awareness ANSR Supra and Sub diaphragmatic and BPQ total score) …”

Page 16: “In these cases we identified a positive significant correlation with BPQ total score and the presence of Cranio-Facial Pathologies …”.

Page 20: “Our findings showed a positive correlation between ANSR Subdiaphragmatic subscale and total score and the use of psychoactive drugs.”

The authors should remove all references to the total score if they truly excluded these analysis from their work.

Response: Thank you for your suggestion. All the sentences referring to the total score have been removed as suggested.

- Binary Scoring System

In my first revisions, I asked why the authors decided to use the binary scoring system for the factor analysis. The authors argued decided that they decide to replicate the previous work. However, I still think the authors should provide alternative results for the full-scoring system, even if these results are displayed in the supplementary materials. The full-scoring system for the BPQ provides much more sensitivity for individual differences and should be used rather than the binary scoring system.

Thus, the authors should at least present an EFA with the full-scoring ratings. This is valuable information for the field as it would allow to understand if in the Italian the full-scoring rating also leads to a large number of factors and loadings with complex factors.

Response: Thank you for your comment. We have addressed this issue.

- Statistical Analysis and Reporting

1) The CFA procedures described in the Methods section is still quite incomplete. Further details should be given to the reader (e.g. type of matrix used, estimation procedure, are factors allowed to correlate or not, etc), at least in Supplemental materials.

2) Correlations with dichotomous variables: Even after the first revisions, the authors still describe and discuss correlations between scores and dichotomous variables like Smoking, Physical Activity or Drug Usage. As I argued before, correlations are not well suited for dichotomous variables. Correlations should be used with continuous variables such as Age or BMI. For dichotomous variables (e.g. Smoking, Physical Activity, Drug Usage, etc), the authors should only report independent samples t-tests comparing both groups (e.g. Drug Users vs. Non-Drug Users).

3) Regression models: I still do not clearly understand which regression models were implemented. More particularly, it is not clear which predictors were used. It seems that age, gender, physical activity, pathologies and drugs were used in the models presented in the supplementary tables. Smoking and BMI were not included in these models, apparently. Why? And psychiatric disorders were not included as well! This predictor is an important clinical outcome. Why was is not included?

Response: Thank you for your suggestion, we conducted and reported the new analysis that you request. 

More importantly, I cannot agree with the authors argument for conducting the models with all the pathologies and drugs as predictors. First, the authors presented linear models with more than 20 predictors. The sample size is not adequate for that many predictors. Secondly, there are predictors which do not have a sufficient amount of subjects per category to allow them to be included in the model.

For instance, there is only 1 subject that reported Immunosuppressor drug use. How can this be a predictor in the model? The same for cancer (n=2) and haematological disorders(n=3)? I do not understand the argument that “used pathologies are considered as an independent factor, which is weighted within the model”. The problem in the analysis is that these models include several dichotomous predictors which lack an adequate number of subjects per category. This is inadequate in my opinion, regardless of the independent factor argument.

4) Reporting of statistical results is not very adequate and does not follow any guidelines. The authors should at least report data for statistical tests using the APA guidelines.

Response: Thank you for your suggestion. To overcome the problem of the appropriateness of models we decided to conduct linear regression using a stepwise method to select predictors. 

- Limitations and Recommendations for future studies

The authors stated that they added a sentence about limitation after each discussion section. However, I still feel that there the manuscript lacks a clear paragraph describing limitations and recommendations for future studies. For instance, the authors do not debate the limitations of their sampling procedures. Collecting data only from subjects looking for osteopathic care is a limitation of this study. Furthermore, this sample is quite well-educated (40% with a university degree), which further limits the conclusions of this work. Also, the authors could also provide clear suggestions for future studies that would allow for convergent validity.

Response: Thank you for your comment. A dedicated section has been added.

- Written Expression and Citations

Written expression is still quite flawed across the document. Several sentences are extremely hard to understand. To be considered for publication, this manuscript should be revised by someone fluent in English writing. There are also still a few issues with citation. For instance, the work form Cabrera et al. (1) is cited across the text in sentences where it is not adequate in my opinion. This should also be addressed.

Response: Thanks for your comment. The manuscript has been revised by native English speakers. 

Introduction

This section has improved significantly since the last review. The constructs (and how they are linked) are more plainly defined. The text is also much more readable and clear for the reader. There is still one minor issue to be addressed.

- Page 3, 2nd paragraph: In my opinion, if body awareness equates to body perception in the authors’ opinion, this concept should be introduced in the first paragraph: “…. Clustered into a construct called body awareness (also known as body perception)”.

Furthermore, the definition of body awareness provided in the 2nd paragraph should be moved somewhere in the first paragraph. It does not make much sense to present the definition only in the 2nd paragraph as the construct was already introduced to the reader in the previous paragraph.

Response: Thank you for your comment. We added the specifications as per your suggestion. We feel that, following the flow of the discourse, there is no need to change the order of the presented topics.

Methods

Page 5

- In authors reply to my original review, they state: “… we have recruited subjects afferent to osteopathic care, and probably that may influence the selection of sample. If it is correct, we should expect a high presence of pathologies related to the osteopathic profession, which affect the musculoskeletal or neurological system.” – I believe this issue should be addressed in the Participants heading.

Response: Thank you for your comment. Only 2% declared to have neurologic disorders, and we have no data concerning the musculoskeletal conditions of participants (which would fall eventually into “others disorders” that account for 5%). So we don’t feel we can support that statement. 

- “The total number of participants was calculated taking into account at least 10 participants for each item of the questionnaire”. Unusual phrasing to explain sample size calculation. Please improve.

Response: Thanks for your comment. The manuscript has been revised by native English speakers. 

- The questionnaires were completed “in the presence of the osteopath” but it still not clear where. Private practices of each osteopath?

Response: Thank you for your comment. “In their private practice” has been added to the sentence.

- “The use of this compilation method …”. Not sure what the authors means by “compilation method”.

Response: Thank you for your comment. The compilation method refers to the google form which was set to “all mandatory responses” in order to avoid missing answers.

Page 6

- Incomplete information on sociodemographic and clinical variables. For instance, regarding physical activity you should at least state that you asked whether subjects performed physical activity ≥2 times/week. Regarding smoking, it should be clear that it was a yes and no question. “Smoking habits” leads readers to wonder whether you asked how many cigarettes per day or per week.

Response: Thank you for your comment. A specification of the timing of physical activity has been added to the table. 1. The “smoker” question did not ask for the number of cigarettes.

- “It has demonstrated strong psychometric properties and a consistent factor structure across multiple languages”. – This is somewhat of an overstatement. To my knowledge, the BPQ was only formally valeted in English and Spanish.

Response: Thank you for your comment. In this website (https://www.stephenporges.com/body-scaleshttps://www.stephenporges.com/body-scales) it’s possible to find all the languages in which the BPQ has been translated.

- The BPQ description is still far too incomplete for a validation paper in my opinion. Although some information can be found in the Results section, I believe that this should be presented together with the scale description. The authors should describe how many items are included in each domain. Also, there is an item that is included in both the supradiaphragmatic reactivity and the subdiaphragmatic reactivity domains. This should be transparent for the reader. It should also be clear that separate scores are completed for the Body Awareness and Autonomic Reactivity domains (the BPQ manual does not even address a total score).

Response: Thank you for your suggestion. Done.

Page 7

- “The process of cross-cultural adaptation followed established guidelines 1.” – This citation does not seem right as the work from Cabrera does not provide any sort of guidelines for cross-cultural adaptation.

Response: Thank you for your suggestion. The sentence has been rephrased. 

Page 8

- Again, citation number 1 (Cabrera et al.) seem to be inadequately used to explain structural validity and construct validity. Please use adequate citations to justify your rationale for psychometric analysis.

Response: Thank you for your comment. As we changed the introduction sentence (previous comment) now it’s clear that we refer to cabrera statistical analysis since we followed the same procedure of that cross-cultural validation of the same questionnaire.

Page 9

- “As suggested by the Consensus-based Standards for the selection of health Measurement Instruments (COSMIN) initiative 23 and the International Society for Quality of Life Research (ISOQOL) 22” – This sentence seems incomplete.

Response: thank you for your comment. The sentence has been rephrased.

- “The hypotheses were that the age was normally distributed with respect to the Awareness subscale values ….” – I could not understand what the authors meant with this sentence.

Response: Thanks for your comment. The manuscript has been revised by native English speakers. 

- “Furthermore, we hypothesized that Smoking habits, BMI and educational level with the Awareness Subscale values.” – This sentence seems incomplete or incorrect.

Response: thank you for your comment. The sentence has been rephrased.

- “CFA was performed considering the subscales of BPQ, the Awareness and ANSR Supra and Sub-diaphragmatic.” – It is not clear that you conducted two separate CFA: 1) a one-factor solution for the Body Awareness domain; 2) a two-factor solution for the Autonomic Reactivity domain. Please improve this. 

Response: Thank you for your comment. The sentence has been improved following your suggestion.

Page 10

- “The Cronbach’s alpha and McDonald’s omega were calculated for the total scale and after removing each item to evaluate this measurement property. We evaluated the internal consistency related to BPQ Awareness, BPQ ANSR total, and in the supra/subdiaphragmatic subscales.” This is extremely confusing. First, the authors state that they computed the internal consistency for the total scale, which does not make much sense as there is no formal recommendation to use the total score for the BPQ. Then, I do not understand what the authors meant when they state “… after removing each item to evaluate this measurement property.”. Please clarify.

Response: Thank you for your comment. The “total score” has been removed and the sentence has been rephrased.

- I think the authors should describe cut-off scores to classify internal consistency scores.

Response:Thank you for your comment. Done.

- “The study of the correlations among BPQ scores and demographic characteristics was performed using Linear Regression Model, and estimating Kendall and Pearson’s indicators.” – Extremely unusual written expression. Please improve.

Response: Thanks for your comment. The manuscript has been revised by native English speakers. 

- “Studying Smoke and Physical activity variables, we computed t-test between smokers and no smokers, and between active and inactive subjects, in order to assess differences in Awareness and ANSR subscales” – Somewhat confusing sentence. Please improve.

Response: Thanks for your comment. The manuscript has been revised by native English speakers. 

- “In the four linear regressions we set, each one with the respective subscales of the BPQ (Awareness ANSR Supra and Sub diaphragmatic and BPQ total score)” – Still wondering why a regression model was computed for the BPQ Total Score. What does this BPQ Total Score measure actually, according to the authors? For me it seems that the BPQ subscales address different constructs (although they can be related with each other). So, if the authors use the BPQ total score they should clearly define what this total score represents.

Response: Thank you for your comment. All the “total score” related sentences have been removed.

“ … as the dependent variables and the demographic and clinical characteristics of the sample as independent variables.” – It is still not clear which variables were included as predictors. This is not clearly described on any part of the manuscript.

Response: Thanlìk you! Done!

Page 11

- “Floor and ceiling effects were considered to be present if ≥ 15% of the patients reported the lowest (0) or highest (47) possible BPQ score” – The BPQ Short Form only was 46 items. So the total score should only be 46, using the binary scoring system. If the authors considered the max score of 47, this is because they probably included item #41 (“I feel like vomiting”) twice, because this item is included on both the Supra and Sub Diaphragmatic domains. This would be incorrect in my opinion. Regardless, I do not think the authors should even consider the total score for analysis. However, if they decide to do so, this issue should be addressed.

Response: Thank you for your comment, we added an explanation.

Results

Page 14

- “The BPQ‐ SF was scored by adding the dichotomized responses (0 = never, 1 = occasionally or more often) in accord to the factor structure described above, with “I feel like vomiting” included in both reactivity scales (supra and subdiaphragmatic subscales).” – In my opinion, this information should be in the Methods section, rather than here.

Response: Thank you for your comment. Following your previous comment we added the information also in the methods.

Page 15, 16 and 17

I still find the description of the results of the section “Relation to demographic variables” quite confusing, for several reasons:

1. Why did you retain the correlation between Physical Activity and Smoking Habits? This variables are dichotomous correct? Thus, only the t-test should be used, not the correlation analysis.

Response: Thank you for your comment. We reorganized this section adding the analysis suggested.

2. The authors stated that they removed the analysis for the Total score in their reply to the first revision. However, they still report the regression model for the total score. Please clarify this.

Response: Thank you for your comment. All the “total score” related sentences have been removed.

3. The results are presented without any clear structure and it very hard to follow them. For instance, I still don’t understand clearly which variables were included in the regression models. In the results section, may be it would be easier to report the correlation, t-tests and ANOVA first. Them, prepare one short paragraph for each regression model, so that the reader can clearly understand the results.

Response:Thank you for your comment. We reorganized this section adding the analysis suggested.

4. Statistical reporting is quite poor. The authors should consider the APA guidelines for reporting their results.

Response:Thank you for your comment. Done.

Page 16

- I still think that Figure 1 should be removed. It’s not useful to interpret the study results.

Response: Thank you for your comment. Done.

Page 17

- “Floor or Ceiling effects were not present in our sample considering the ≥ 15% of the patients

reported the lowest (0) or highest (47) possible BPQ score” – Improve written expression.

Response: Thanks for your comment. The manuscript has been revised by native English speakers. 

Discussion

Page 18

- “For these reasons, we argue that the results of the MIA influence only partially the cross-cultural adaptation and the validity of the Italian version of the BPQ-SF” – Improve written expression

Response: Thanks for your comment. The manuscript has been revised by native English speakers. 

- “The internal consistency showed to be very high for both subscales of the BPQ, and for the two subscales of ANSR (Table 5).” – To make this statement the authors should have defined cut-off criteria for the internal consistency measures. Please address this.

Response: Thank you. Done.

- “Moreover, since it was not planned a convergent/discriminant validity testing it might be difficult to assess to what extent the Italian version of the BPQ measures body awareness and automatic reactivity” After this statement, I think the authors should address how this is particularly true for the BPQ Body Awareness subscale and there is a lot of debate regarding the different measurement models of interception. I know the authors introduce this issue subsequently, but this seems the ideal spot to introduce this issue.

Response: Thank you for your comment. We feel that, following the flow of the discourse, there is no need to change the order of the presented topics

Page 19

- “However, there is some evidence in contrast with the negative association we observed between physical activity and ANSR 37. Physical activity correlated negatively with the subscale Awareness, which is not consistent with other studies using different instruments of body awareness 37,38” – Two sentences addressing the same argument. Please revise.

Response: Thank you for your comment. Done.

- “In fact, Multidimensional Assessment of Interoceptive Awareness which is a questionnaire that measures interoception with a very different framework from the BPQ.” – This sentence is incomplete.

Response: Thank you for your comment. The sentence has been rephrased.

- “Interestingly being a smoker seems to correlate with a heightened awareness on the BPQ awareness subscale.” – Correlate may not be the most adequate term. Also written expression should be improved in this sentence.

Response: Thank you for your comment. Done.

- “ … the interpretation framework between the awareness of the interoception and smoking takes into account that the insula - which can be affected by nicotine making the interoceptive information available to conscious awareness.” – After reading it multiple times, I still cannot understand this sentence. Please revise it.

Response: Thanks for your comment. The manuscript has been revised by native English speakers. 

Page 20

- “This result might be correlated with the findings from the study of Danner et al 46 in which the highest awareness …” – Again, correlated is not the most adequate term.

Response: Thank you for your comment. Done.

- “Our findings showed a positive correlation between ANSR Subdiaphragmatic subscale and total score and the use of psychoactive drugs. This finding is in line with previous studies 49” – Again, the authors stated in the reply to the revision that the total score would be removed, however they still debate it in the Discussion section. Furthermore, I do not understand how can the authors report a correlation with the use of psychoactive drugs, which is (if I understood correctly) a dichotomous variable. At the most the authors could compare drug users and non-users with an independent samples t-test.

Response: Thank you, done.

Sincerely,

The authors

---

## [Decision Letter · Decision Letter 2]

4 Jan 2021

PONE-D-19-35072R2

Cross-cultural adaptation and psychometric properties of the Italian version of the Body Perception Questionnaire.

PLOS ONE

Dear Dr. Consorti,

Thank you for submitting your manuscript to PLOS ONE. After careful consideration, we feel that it has merit but does not fully meet PLOS ONE’s publication criteria as it currently stands. Therefore, we invite you to submit a revised version of the manuscript that addresses the points raised during the review process.

We look forward to receiving your revised manuscript.

Kind regards,

Antonio Palazón-Bru, PhD

Academic Editor

PLOS ONE

Reviewers' comments:

Reviewer's Responses to Questions

**Comments to the Author**

1. If the authors have adequately addressed your comments raised in a previous round of review and you feel that this manuscript is now acceptable for publication, you may indicate that here to bypass the “Comments to the Author” section, enter your conflict of interest statement in the “Confidential to Editor” section, and submit your "Accept" recommendation.

Reviewer #1: (No Response)

2. Is the manuscript technically sound, and do the data support the conclusions?

Reviewer #1: Partly

3. Has the statistical analysis been performed appropriately and rigorously? 

Reviewer #1: No

4. Have the authors made all data underlying the findings in their manuscript fully available?

Reviewer #1: No

5. Is the manuscript presented in an intelligible fashion and written in standard English?

Reviewer #1: Yes

6. Review Comments to the Author

Reviewer #1: The manuscript was fairly improved, as the authors completed most request minor changes. However, there are still major issues that should be addressed. Statistical reporting is still a major issue that needs to be improved so that the reader can clearly understand the psychometric properties of this questionnaire. Also, although the findings presented here partially support the BPQ-I as an effective tool to measure body awareness and autonomic reactivity, I believe that the current work has several limitations that are not clearly highlighted by the authors. This is extremely important to the field as there is a lot of debate about whether these measures are truly measuring what they are intended to measure. For instance, in the Abstract the authors argue: “Our results establish the BPQ-I for measurement of body awareness and experiences of autonomic reactivity” and this seems somewhat of an overstatement. Bellow, I provide my third round of revisions hoping that the authors can place their efforts to majorly improve the manuscript so that this relevant work can be published according to PLoS One standards.

Exploratory Factor Analysis: As suggested in my previous revisions, the authors added the exploratory factor analysis using the full-scoring system of the BPQ. This is extremely relevant when debating whether the BPQ is suitable to assess body awareness and autonomic reactivity. However, there are still several issues regarding this topic.

1. The authors state “… we computed an EFA considering the same subscales and factors structure of CFA analysis”. This is quite confusing, as I do not properly understand the statistical procedures implemented for EFA analysis. Much more information should be provided to the reader. Also, using a priori factor structure for an exploratory analysis is not very advisable. The authors should clarify how did they implement the EFA.

2. The authors should also describe the importance of conducting an EFA for the full-scoring system in the Methods section, highlighting the problematic results reported by Cabrera et al. when they conducted a similar analysis.

3. The authors provided very limited results regarding the EFA and this is quite problematic. They should present to the reader which solutions were retrieved from these analyses, namely the number of factors and respective loadings. This would be critical to understanding whether the “untenably high numbers of factors and loadings with complex structure” found by Cabrera et al. was also an issue in this sample. Reporting these results clearly and transparently would be critical to researchers using these measures.

4. The EFA results for the full-scoring system should be ideally reported before the CFA for the binary scoring system. For instance, there would be no reason to use the binary scoring system if the EFA results provided a suitable solution for the full-scoring system, as the latter would provide much more sensitivity for individual differences.

Confirmatory Factor Analysis: The authors did not make any adjustment regarding this previous comment: “The CFA procedures described in the Methods section are still quite incomplete. Further details should be given to the reader (e.g. type of matrix used, estimation procedure, are factors allowed to correlate or not, etc), at least in Supplemental materials.”

Regression Analysis: The authors completed several adjustments to the regression models. However, there are still a few issues that cannot be overlooked. For instance, why was physical activity only included as a predictor on one of the models? Also, more importantly, the authors still included predictors that are not suitable for analysis. There is still only 1 subject that reported Immunosuppressor drug use. This is not a valid variable to be included in a regression model and using stepwise regression models does not address this issue.

Limitations, Recommendations for future studies & Convergent Validity: The authors added limitations and future studies heading, but the information provided there seems scarce. First, they should start that section by highlight the major limitation of the study: the lack of convergent and divergent validity. The results presented by the authors do not allow to affirm that BPQ-I is measuring body awareness and autonomic reactivity. This is particularly true regarding the body awareness subscale, as interoception-related measures have been widely questioned in the field. Thus, authors should clearly highlight this and recommend future studies to further explore this issue. Second, they describe that the current sample had high education levels, but they do not discuss the implications of this (e.g., is body-related phenom easily interpreted by participants with less education). Finally, the limitation regarding the EFA results was not clear to me, probably because the EFA implementation and results were not clearly described earlier.

Participants: In the first round of revisions, the authors argued: “… we have recruited subjects afferent to osteopathic care, and probably that may influence the selection of sample. If it is correct, we should expect a high presence of pathologies related to the osteopathic profession, which affect the musculoskeletal or neurological system.” In this second round of revision, the authors state: “ … Only 2% declared to have neurologic disorders, and we have no data concerning the musculoskeletal conditions of participants (which would fall eventually into “others disorders” that account for 5%). So we don’t feel we can support that statement.” This is quite contradicting. But mainly, the point of my revision regarding participant recruitment is that the current sample is not a regular community sample, but rather an osteopathic care sample, where a significant % of subjects report some sort of health-related condition as described in the Supplementary materials. This is a significant limitation of this study and should be clear for the reader in the Participants heading.

Internal consistency interpretation: The values retrieved from Cabrera et al. should no be used to define criteria for internal consistency interpretation. Although not consensual, there are guidelines to interpret internal consistency values.

Internal consistency heading: The authors should make it clear that internal consistency was computed for each subscale separately, as requested in the previous revision.

Floor and ceiling effects: As presented in the previous revision, I do not believe that item 47 should be considered twice when assessing floor and ceiling effects.

Incomplete information on sociodemographic and clinical variables: The author state “A specification of the timing of physical activity has been added to the table. 1. The “smoker” question did not ask for the number of cigarettes.” This should be clear not only in the Table but also when describing the online survey.

BPQ Description: “The BPQ description is still far too incomplete for a validation paper in my opinion. Although some information can be found in the Results section, I believe that this should be presented together with the scale description. The authors should describe how many items are included in each domain. Also, there is an item that is included in both the supradiaphragmatic reactivity and the subdiaphragmatic reactivity domains. This should be transparent for the reader. It should also be clear that separate scores are completed for the Body Awareness and Autonomic Reactivity domains (the BPQ manual does not even address a total score).” I believe that this comment, provided in the previous revision, was not still adequately addressed by the authors.

Cabrera et al. Citation: After reading the manuscript, I still feel that the Cabrera et al. citation is misused across the document. It seems that the rationale from some analytical methods was developed within this work which, sometimes, is clearly not the case. For instance, “based on the assumption that the instrument validly measures the construct to be measured1” (Page 8).

Page 8 & 9: “The hypotheses were that the age was normally distributed with respect 9 to the Awareness subscale values and that age was negatively correlated with ANSR subscale value.” I still do not understand this hypothesis. Please clarify.

Page 8: “Consensus-based Standards for the selection of health Measurement Instruments (COSMIN) initiative 23 and the International Society for Quality of Life Research (ISOQOL) 22 have been used as conceptual framework”. I believe this information should be provided at the beginning of the heading.

Discussion about smoking: This paragraph needs to be revised. Although there were no differences between groups regarding the BPQ, the authors state that “the observed correlation converges with prior evidence and theory.”

Statistical Reporting: There are still fairly unusual issues regarding statistical reporting For instance, using “p-value < 0.05” instead of simply using “p < 0.05”. Also, the authors should provide the exact p-value whenever possible.

Written expression: Written expression was widely improved across the whole document but there are still some issues that need to be revised. Bellow, there are some of these issues which I picked up.

Page 3

Original: “ … transmitted to the brain forming a neural pathway through …”

Suggested revision: “ … transmitted to the brain, forming a neural pathway through …”

Page 4

Original: “important sources of information that patient and clinician”

Suggested revision: “important sources of information that patients and clinicians”

Original: “trough efferent nerves that originate in the nucleus ambiguus in the brainstem”

Suggested revision: “trough efferent nerves that originate from the nucleus ambiguous in the brainstem”

Page 12

“Extensive information on the CFA is reported in Table 3.” Table 3 reports results from Measurement Invariance Analysis

Page 15

Original: “in accord to the factor structure described above”

Suggested revision: “in accordance to the factor structure described above”

Page 16

“Participants who were physically active were [HIGHER? LOWER?] than those who were inactive” - ???

Page 18

“Negative associations were found between ANSR subscale and age, physical activity, and

male gender, as well as and between age and male gender in the awareness subscale” – Rephrase this sentence

7. PLOS authors have the option to publish the peer review history of their article (what does this mean?). If published, this will include your full peer review and any attached files.

Reviewer #1: No

---

## [Author Response · Author response to Decision Letter 2]

8 Mar 2021

Review of the manuscript

Manuscript number PONE-D-19-35072, entitled “Cross-cultural adaptation and validity of the Italian version of the Body Perception Questionnaire.”

Dear reviewers,

We greatly appreciate your detailed and thoughtful suggestions on the manuscript, that we acknowledge have improved the quality of the manuscript. A detailed description of all changes has been provided below.

For any further information, please do not hesitate to contact us.

Reviewer 1

Reviewer #1: The manuscript was fairly improved, as the authors completed most request minor changes. However, there are still major issues that should be addressed. Statistical reporting is still a major issue that needs to be improved so that the reader can clearly understand the psychometric properties of this questionnaire. Also, although the findings presented here partially support the BPQ-I as an effective tool to measure body awareness and autonomic reactivity, I believe that the current work has several limitations that are not clearly highlighted by the authors. This is extremely important to the field as there is a lot of debate about whether these measures are truly measuring what they are intended to measure. For instance, in the Abstract the authors argue: “Our results establish the BPQ-I for measurement of body awareness and experiences of autonomic reactivity” and this seems somewhat of an overstatement. Bellow, I provide my third round of revisions hoping that the authors can place their efforts to majorly improve the manuscript so that this relevant work can be published according to PLoS One standards.

Exploratory Factor Analysis: As suggested in my previous revisions, the authors added the exploratory factor analysis using the full-scoring system of the BPQ. This is extremely relevant when debating whether the BPQ is suitable to assess body awareness and autonomic reactivity. However, there are still several issues regarding this topic.

Response: Thank you for your work, we really appreciate your dedication and prior recommendations. The paper now has been revised to better match your requests and we do acknowledge that these have made for an improved manuscript. 

1. The authors state “… we computed an EFA considering the same subscales and factors structure of CFA analysis”. This is quite confusing, as I do not properly understand the statistical procedures implemented for EFA analysis. Much more information should be provided to the reader. Also, using a priori factor structure for an exploratory analysis is not very advisable. The authors should clarify how did they implement the EFA.

Response: While the initial plan for the analysis was to conduct a confirmatory factor analysis, we do agree that an exploratory factor analysis can provide information on alternate and optimal structure to the BPQ. We have now included comprehensive a detailed description of the EFA in the supplemental material, and these results are referenced in the main text.

2. The authors should also describe the importance of conducting an EFA for the full-scoring system in the Methods section, highlighting the problematic results reported by Cabrera et al. when they conducted a similar analysis.

Response: 

3. The authors provided very limited results regarding the EFA and this is quite problematic. They should present to the reader which solutions were retrieved from these analyses, namely the number of factors and respective loadings. This would be critical to understanding whether the “untenably high numbers of factors and loadings with complex structure” found by Cabrera et al. was also an issue in this sample. Reporting these results clearly and transparently would be critical to researchers using these measures.

Response: We have now included a full description of the exploratory factor analysis conducted on the full-item distributions, which has now been included in the supplemental materials.

4. The EFA results for the full-scoring system should be ideally reported before the CFA for the binary scoring system. For instance, there would be no reason to use the binary scoring system if the EFA results provided a suitable solution for the full-scoring system, as the latter would provide much more sensitivity for individual differences.

Response: We agree with the reviewer that the EFA results may supersede the CFA. However, it is not advisable to conduct EFA and CFA on the same sample and our sample was not large enough to split into two sections. Based on our review of the literature and prior studies, as well as goals of conducting the CFA so that measurement invariance could be assessed (a goal that cannot be completed with EFA), the EFA must be considered a post-hoc analysis to test whether alternate factor structures are more plausible. Thus, the CFA is reported in the main section of the manuscript and the EFA in the supplemental material. 

5. Confirmatory Factor Analysis: The authors did not make any adjustment regarding this previous comment: “The CFA procedures described in the Methods section are still quite incomplete. Further details should be given to the reader (e.g. type of matrix used, estimation procedure, are factors allowed to correlate or not, etc), at least in Supplement materials.”

Response: Thank you, we specified better all the features requested in the first paragraph of Factor Analysis and Measurement Invariance Analysis (MIA) (pag.10)and we added the factor loadings in S5 Table.

6. Regression Analysis: The authors completed several adjustments to the regression models. However, there are still a few issues that cannot be overlooked. For instance, why was physical activity only included as a predictor on one of the models? Also, more importantly, the authors still included predictors that are not suitable for analysis. There is still only 1 subject that reported Immunosuppressor drug use. This is not a valid variable to be included in a regression model and using stepwise regression models does not address this issue.

Response: Thank you for your suggestion. In the last manuscript we use a backward stepwise regression to select variables as predictors in our model. As you have well pointed out in some models physical activity was not included and other problems, as Immunosuppressant drug use, were still present. In order to improve our paper, we establish a-priori model that also includes the “physical activity” variable and we dichotomised the variable “use of drugs” in: medications use yes/no. We decided this strategy, as you suggest, considering the distribution of the responses. 

7. Limitations, Recommendations for future studies & Convergent Validity: The authors added limitations and future studies heading, but the information provided there seems scarce. First, they should start that section by highlight the major limitation of the study: the lack of convergent and divergent validity. The results presented by the authors do not allow to affirm that BPQ-I is measuring body awareness and autonomic reactivity. This is particularly true regarding the body awareness subscale, as interoception-related measures have been widely questioned in the field. Thus, authors should clearly highlight this and recommend future studies to further explore this issue. Second, they describe that the current sample had high education levels, but they do not discuss the implications of this (e.g., is body-related phenom easily interpreted by participants with less education). Finally, the limitation regarding the EFA results was not clear to me, probably because the EFA implementation and results were not clearly described earlier.

Response: Thank you for your comment. We added an highlight of this issue in the limitation section as follows “The present study includes some limitations. The psychometric evaluation of the questionnaire did not encompass the assessment of convergent and divergent validity. Therefore, it is not possible to affirm that the BPQ-I actually measures the intended constructs (i.e. body awareness and autonomic reactivity). Although the body awareness construct has been studied in previous studies [49], the autonomic reactivity construct still needs to be tested with sensor-based measures. Further studies are needed to better define those constructs and their properties. Participants were recruited from a pool of clients seeking osteopathic care. Therefore, the current sample is not a regular community sample, but rather an osteopathic care sample, where a significant percentage of subjects reported a health-related condition. It is possible that participants in this sample may have higher autonomic reactivity or lower awareness as part of their reason for seeking care. Further studies need to be conducted with more general samples to establish better normative data and replicate the psychometric features of the scale. Furthermore, the participants in the sample were highly educated (40% with a university degree). A previous study underlined how education level might impact the body awareness of subjects with a direct correlation [54]. However, the sample in the study was specific and therefore the results might be difficult to generalize. In order to understand the relationship between education level and body awareness further studies are needed.”

8. Participants: In the first round of revisions, the authors argued: “… we have recruited subjects afferent to osteopathic care, and probably that may influence the selection of sample. If it is correct, we should expect a high presence of pathologies related to the osteopathic profession, which affect the musculoskeletal or neurological system.” In this second round of revision, the authors state: “ … Only 2% declared to have neurologic disorders, and we have no data concerning the musculoskeletal conditions of participants (which would fall eventually into “others disorders” that account for 5%). So we don’t feel we can support that statement.” This is quite contradicting. But mainly, the point of my revision regarding participant recruitment is that the current sample is not a regular community sample, but rather an osteopathic care sample, where a significant % of subjects report some sort of health-related condition as described in the Supplementary materials. This is a significant limitation of this study and should be clear for the reader in the Participants heading.

Response: Thank you for your comment. We added a sentence to specify that both in the “participants” paragraph and in the “limitations” paragraph. Respectively: “Therefore, the current sample might not be considered a regular community sample, but rather an osteopathic care sample.” and “Therefore, the current sample is not a regular community sample, but rather an osteopathic care sample, where a significant percentage of subjects reported a health-related condition”

9. Internal consistency interpretation: The values retrieved from Cabrera et al. should no be used to define criteria for internal consistency interpretation. Although not consensual, there are guidelines to interpret internal consistency values.

Response: Thank you for your comment. The mentioned section has been removed.

10. Internal consistency heading: The authors should make it clear that internal consistency was computed for each subscale separately, as requested in the previous revision.

Response: Thank you for your comment. We added the following sentence “Internal consistency was computed for each subscale separately”

11. Floor and ceiling effects: As presented in the previous revision, I do not believe that item 47 should be considered twice when assessing floor and ceiling effects.

Response: Thank you, we reconsider the items (46).

12. Incomplete information on sociodemographic and clinical variables: The author state “A specification of the timing of physical activity has been added to the table. 1. The “smoker” question did not ask for the number of cigarettes.” This should be clear not only in the Table but also when describing the online survey.

Response: Thank you for your comment. We included a better description of the questions in the proper online survey section: “Questions concerning medications use, physical activity and smoking habits allowed only dichotomous answers (yes/no) without quantitative specification, apart from the physical activity question (if ≥2/week)”

13. BPQ Description: “The BPQ description is still far too incomplete for a validation paper in my opinion. Although some information can be found in the Results section, I believe that this should be presented together with the scale description. The authors should describe how many items are included in each domain. Also, there is an item that is included in both the supradiaphragmatic reactivity and the subdiaphragmatic reactivity domains. This should be transparent for the reader. It should also be clear that separate scores are completed for the Body Awareness and Autonomic Reactivity domains (the BPQ manual does not even address a total score).” I believe that this comment, provided in the previous revision, was not still adequately addressed by the authors.

Response: Thank you for your comment! We created a specific section for BPQ description.

14. Cabrera et al. Citation: After reading the manuscript, I still feel that the Cabrera et al. citation is misused across the document. It seems that the rationale from some analytical methods was developed within this work which, sometimes, is clearly not the case. For instance, “based on the assumption that the instrument validly measures the construct to be measured1” (Page 8).

Response: Thank you for your comment. The assumption described in that sentence refers to the fact that Cabrera et al. studied the construct validity through factorial analysis. Indeed, the sentence specifically referred to the english and spanish version of the BPQ. Anyway, following your appropriate suggestion, all the references about Cabrera et al. were reviewed and where pertinent were changed accordingly.

15. Page 8 & 9: “The hypotheses were that the age was normally distributed with respect 9 to the Awareness subscale values and that age was negatively correlated with ANSR subscale value.” I still do not understand this hypothesis. Please clarify.

Response: Thank you for your comment. The hypotheses arose from previous published works. We added references for each hypothesis.

16. Page 8: “Consensus-based Standards for the selection of health Measurement Instruments (COSMIN) initiative 23 and the International Society for Quality of Life Research (ISOQOL) 22 have been used as conceptual framework”. I believe this information should be provided at the beginning of the heading.

Response: Thank you for your comment. Done.

17. Discussion about smoking: This paragraph needs to be revised. Although there were no differences between groups regarding the BPQ, the authors state that “the observed correlation converges with prior evidence and theory.”

Response:Thank you for your comment. The sentence has been changed as follow “the observed correlation did not converges with prior evidence and theory”

18. Statistical Reporting: There are still fairly unusual issues regarding statistical reporting For instance, using “p-value < 0.05” instead of simply using “p < 0.05”. Also, the authors should provide the exact p-value whenever possible.

Response: Thank you! Done!

19. Written expression: Written expression was widely improved across the whole document but there are still some issues that need to be revised. Bellow, there are some of these issues which I picked up.

Response: Thanks

20. Page 3

Original: “ … transmitted to the brain forming a neural pathway through …”

Suggested revision: “ … transmitted to the brain, forming a neural pathway through …”

Response: Thank you. Done.

21. Page 4

Original: “important sources of information that patient and clinician”

Suggested revision: “important sources of information that patients and clinicians”

Response: Thank you. Done.

22. Original: “trough efferent nerves that originate in the nucleus ambiguus in the brainstem”

Suggested revision: “trough efferent nerves that originate from the nucleus ambiguous in the brainstem”

Response: Thank you. Done.

23. Page 12

“Extensive information on the CFA is reported in Table 3.” Table 3 reports results from Measurement Invariance Analysis

Response: Thank you. Done.

24. Page 15

Original: “in accord to the factor structure described above”

Suggested revision: “in accordance to the factor structure described above”

Response: Thank you. Done.

25. Page 16

“Participants who were physically active were [HIGHER? LOWER?] than those who were inactive” - ???

Response: thank you, done!

Page 18

“Negative associations were found between ANSR subscale and age, physical activity, and

male gender, as well as and between age and male gender in the awareness subscale” – Rephrase this sentence

Response: Thank you for your comment. The sentence has been rephrased as follows “Negative associations were found between ANSR subscale and age, physical activity, and male gender. Similarly it was observed a negative association between age and male gender and the awareness subscale.”

We would like to sincerely thank the reviewer for the incredibly accurate revision of the manuscript that radically improved the overall quality of the work.

The authors

---

## [Editor Report · Decision Letter 3]

5 May 2021

Cross-cultural adaptation and psychometric properties of the Italian version of the Body Perception Questionnaire.

PONE-D-19-35072R3

Dear Dr. Consorti,

We’re pleased to inform you that your manuscript has been judged scientifically suitable for publication and will be formally accepted for publication once it meets all outstanding technical requirements.

Kind regards,

Antonio Palazón-Bru, PhD

Academic Editor

PLOS ONE

Additional Editor Comments (optional):

All the reviewers' concerns have been correctly addressed.
---

## [Editor Report · Acceptance letter]

11 May 2021

PONE-D-19-35072R3 

Cross-cultural adaptation and psychometric properties of the Italian version of the Body Perception Questionnaire. 

Dear Dr. Consorti:

I'm pleased to inform you that your manuscript has been deemed suitable for publication in PLOS ONE. Congratulations! Your manuscript is now with our production department. 

Kind regards, 

on behalf of

Dr. Antonio Palazón-Bru 

Academic Editor

PLOS ONE